# Antimicrobial Activity Enhancers: Towards Smart Delivery of Antimicrobial Agents

**DOI:** 10.3390/antibiotics11030412

**Published:** 2022-03-18

**Authors:** Mariusz Skwarczynski, Sahra Bashiri, Ye Yuan, Zyta M. Ziora, Osama Nabil, Keita Masuda, Mattaka Khongkow, Natchanon Rimsueb, Horacio Cabral, Uracha Ruktanonchai, Mark A. T. Blaskovich, Istvan Toth

**Affiliations:** 1School of Chemistry and Molecular Bioscience, The University of Queensland, Brisbane, QLD 4072, Australia; m.skwarczynski@uq.edu.au (M.S.); s.bashiri@uq.edu.au (S.B.); 2Centre for Superbug Solutions, Institute for Molecular Bioscience, The University of Queensland, Brisbane, QLD 4072, Australia; ye.yuan1@uq.net.au (Y.Y.); z.ziora@imb.uq.edu.au (Z.M.Z.); m.blaskovich@imb.uq.edu.au (M.A.T.B.); 3Department of Bioengineering, Graduate School of Engineering, The University of Tokyo, 7-3-1 Hongo, Bunkyo-ku, Tokyo 113-8656, Japan; osama-nabil@g.ecc.u-tokyo.ac.jp (O.N.); masudakeita@g.ecc.u-tokyo.ac.jp (K.M.); horacio@bmw.t.u-tokyo.ac.jp (H.C.); 4National Nanotechnology Center (NANOTEC), National Science and Technology Development Agency (NSTDA), 111 Thailand Science Park, Phahonyothin Road, Klong 1, Klong Luang 12120, Pathumthani, Thailand; mattaka@nanotec.or.th (M.K.); natchanon.rim@nanotec.or.th (N.R.); uracha@nanotec.or.th (U.R.); 5School of Pharmacy, The University of Queensland, Brisbane, QLD 4072, Australia

**Keywords:** antibiotic, antimicrobial resistance, delivery systems, activity enhancers, silver, polymers, lipids

## Abstract

The development of effective treatments against infectious diseases is an extensive and ongoing process due to the rapid adaptation of bacteria to antibiotic-based therapies. However, appropriately designed activity enhancers, including antibiotic delivery systems, can increase the effectiveness of current antibiotics, overcoming antimicrobial resistance and decreasing the chance of contributing to further bacterial resistance. The activity/delivery enhancers improve drug absorption, allow targeted antibiotic delivery, improve their tissue and biofilm penetration and reduce side effects. This review provides insights into various antibiotic activity enhancers, including polymer, lipid, and silver-based systems, designed to reduce the adverse effects of antibiotics and improve formulation stability and efficacy against multidrug-resistant bacteria.

## 1. Introduction

Infection-related illnesses are caused by microorganisms, including viruses, bacteria, fungi, and parasites. Among these, pathogenic bacteria are a leading public health concern due to the high morbidity and mortality associated with infections, advanced transmission, and difficulty in managing complex infections [1,2]. The first “modern-day antimicrobial agent”, penicillin, was discovered by Alexander Fleming in 1928 [3]. The following 20 years were known as the “golden era of antibiotics” due to the discovery of a series of other antimicrobial agents, such as sulfonamides and aminoglycosides [3,4]. These compounds proved revolutionary in the treatment of historically fatal infections. Today, antibiotics are widely used to control infectious diseases; however, their over-use in the past century, due to prescription for non-relevant infections and prophylactic use in agriculture, has driven the natural selection of bacteria that are resistant to increasing numbers of antibiotics (AMR) [5,6,7]. In addition, the limited efficacy lifespan of novel antibiotics and competition from cheap generics has made their development less attractive to pharmaceutical companies [6]. Bacterial strains simultaneously resistant to multiple antibiotics have now emerged, resulting in fatalities from previously treatable bacterial infections [4]. Compounded by antibiotic resistance, bacterial infections are currently the second-leading cause of death globally [8]. A 2022 report determined that in 2019 there were nearly 5 million global deaths associated with resistant bacterial infections [9].

### 1.1. Antimicrobial Resistance Mechanism

In principle, resistance can occur through the innate ability of bacteria to survive antibiotic treatment, known as intrinsic resistance mechanisms [10], due to: (a) reduction in drug permeability/uptake; (b) biofilm formation, which reduces the susceptibility of individual bacteria to antibiotic activity; (c) inactivation of antibiotics; (d) enzymatic degradation; and (e) overexpression of efflux pump proteins [4,11,12] (Figure 1). Besides intrinsic resistance, acquired resistance often occurs, due to: (a) drug target mutation; (b) horizontal gene transfer between species by transformation, transduction, or conjugation [13].

In addition to the intrinsic resistance mechanisms common to Gram-negative and Gram-positive bacteria, the outer membrane of Gram-negative bacteria provides another barrier that blocks antibiotic penetrability [11,14,15]. The development of new antibiotics with broad mechanisms of action or utilizing delivery systems to improve the efficacy of existing antimicrobial agents are the main approaches to fight AMR [16]. In this respect, new hope comes for the “post-antibiotic era” to overcome AMR.

### 1.2. Modern Antimicrobial Agents

New therapeutic agents against pathogenic bacteria must be effective against multidrug-resistant (MDR) bacteria and have low potential to develop resistance. Antimicrobial agents are divided into natural agents and synthetic compounds. Natural antimicrobial agents are cultivated from living organisms, such as filamentous saprophytic microbes, or extracted from medicinal herbs, such as curcumin extracted from Zingiberaceae [7,17,18]. Bacteriophages—DNA/RNA viruses that infect bacteria—can also be considered as nature-based antibacterial agents [19]. Synthetic antimicrobial compounds, produced using fully synthetic chemical processes or synthetic biology and genetic engineering, are mostly in the early stages of development—almost all classes of modern antibiotics are derived from natural products.

Furthermore, some classes of antimicrobial agents, for example, antimicrobial peptides (AMPs), can be both nature-derived as well as fully synthetic. AMPs are often isolated from microorganisms (e.g., protozoa, fungi, and bacteria), plants, and animals, including humans. For example, bacteriocin, ribosomal AMPs synthesized by a strain of *Pediococcusacidilactici*, has a natural source; however, GLK-19 is a synthetic peptide with broad-spectrum antimicrobial activity against *Escherichia coli* and methicillin-resistant *Staphyloccocus aureus* (MRSA) USA-300 [20,21,22,23]. AMP sequences consist of 5–50 amino acids, usually with l-configuration, and are often in tridimensional structures: α-helices, β-sheets, or both [24]. AMPs are attractive antimicrobial agents as they have high efficiency, rapid sterilization, small molecular weight (~500–5000 Dalton), appropriate thermal stability, and low immunogenicity, but despite these advantages new AMPs have not succeeded in becoming approved antibiotics [25].

Most AMPs have net positive charges to adhere and interact with the negative bacterial cell surface membrane. This interaction leads to cell membrane rupture, destruction of membrane integrity and, ultimately, cell death. However, this membrane activity often also causes collateral damage to mammalian cells, and peptidic antimicrobial agents can be easily degraded, bind to serum proteins because of their cationic and amphiphilic nature, and are rapidly cleared from bloodstream circulation [20,21,26,27]. Thus, a suboptimal concentration of the drug reaches the bacteria resulting in poor clinical outcomes. High drug doses can overcome this drawback, but overdosing imposes potential toxicity for humans or animals, limiting clinical applicability. Accounting for these deficiencies, several strategies have been developed to enhance the transport of antibiotics to infection sites and overcome the microbial envelope barrier [28]. Delivery systems can also be used to release antibiotics selectively in the phagocytic cells responsible for bacterial clearance [29,30,31,32].

### 1.3. Delivery Systems

Delivery systems are usually designed to improve drug bioavailability, overcome the poor aqueous solubility of hydrophobic drugs, protect cargo from extracellular degradation, prolong systemic drug circulation by increasing antibiotic half-life in plasma, allow sustained antibiotic release, and provide site-specific targeting [33,34,35]. Delivery systems can also minimize antibiotic accumulation in healthy host tissue and simultaneously reduce toxicity [35,36]. Ultimately, delivery systems often act as antimicrobial activity enhancers. Delivery system efficacy strongly depends on the strategy used to incorporate drugs. Chemical conjugation allows the formation of stable conjugates; however, the biological activity of antimicrobial agents can be reduced as a result of the chemical bonds between the drug and carrier. These conjugates may also suffer from inefficient drug release [37]. In contrast, physical entrapment of antimicrobial agents within the carrier protects their biological activity. However, the lack of chemical conjugation between antimicrobial agent and carriers may allow premature drug release. Therefore, careful optimization of a delivery system is required.

Nanosized drug delivery systems (e.g., nanoparticles, nanocarriers) can provide a flexible platform for transporting antimicrobial agents and delivering them to targeted tissues. Over the past five decades, several types of nanoscale delivery vehicles have been explored to encapsulate biomolecules [37,38]. Based on the building material, these delivery systems may be categorized into lipid-, polymer-, or metal-based nanoparticles, among others.

The combination of nanoscale delivery system with antibiotics, known as “nanobiotics”, provides a high surface-to-volume ratio and the possibility of surface functionalization, contributing to high bacterial affinity and efficient antibacterial activity [39]. The efficient transport of nanoparticles through the endothelium of the inflammatory site means that lower drug doses can be administered [40]. The high permeability of the nanoscale delivery system through biological membranes greatly improves antibiotics’ efficacy against intracellular pathogens [8]. In addition, the surface of such delivery systems can be modified with ligands of tissue-specific receptors to allow encapsulated drugs to be transported to desired sites of action and improve cellular uptake [8,35]. 

### 1.4. Nanomaterials in Antibiotic Delivery

Nanomaterials can have antibacterial activity when they are designed as antimicrobial agents or as carriers to encapsulate antibiotics. For example, metal nanoparticles can alter the metabolic activity of bacteria, such as Gram-positive (*Staphylococcus aureus*) and Gram-negative (*Escherichia coli*) species, enhancing the antimicrobial activity of antibiotics [41]. Nanoparticles can tackle multiple biological pathways (Figure 2). Thus, the materials used for antibiotic delivery should also be tested for antimicrobial activity.

Due to their small size, nano delivery system vesicles can deliver antibiotics into cells. In doing so, they are able to fight intracellular pathogens by inhibiting enzyme activity and DNA synthesis, deactivating protein, and inducing oxidative stress and electrolyte imbalance. The induction of oxidative stress by reactive oxygen species (ROS) is a key aspect of nanoparticle action against bacteria. ROS, such as superoxide anion, hydrogen peroxide, and hydroxyl radicles, damage vital components of the pathogen, including membranes, lipids, and proteins. ROS inhibit the replication of DNA, amino acid synthesis, and damage microbial cell membrane [42]. For example, silver nanoparticles induced the formation of hydroxyl radicals in Gram-negative bacteria such as *Pseudomonas aeruginosa* and *Shigella flexneri*; and Gram-positive bacteria including *Staphylococcus aureus* and *Streptococcus pneumoniae* while iron oxide nanoparticles triggered the formation of hydroxyl radical and hydrogen peroxide against *Bacillus subtilis* and *Escherichia coli* [43,44]. Nanoparticles can also interfere with a microbe’s membrane shape and function by disrupting the metabolic pathways responsible for membrane synthesis [45,46]. Furthermore, nanoparticles (e.g., metal nanoparticles, carbon-based nanoparticles) can enter biofilms and prevent biofilm development by suppressing gene expression [47].

The surface of delivery systems can be readily functionalized with ligands, such as antibodies, peptides, and saccharides, directed to target bacteria [48,49,50]. The ligands can promote binding to pathogenic bacteria for maximized efficacy, while minimizing toxicity to beneficial bacteria and mammalian cells. Moreover, antibiotics that bind to bacterial membranes can be used as ligands on the surface of nanocarriers. Besides enhancing selectivity, this approach can augment target affinity through multivalent interactions and promote antibiotic action through high, localized drug density [51,52]. For example, vancomycin ligands on nanoparticles improved the efficacy of antibiotics (15- to >100-fold stronger potency), even against vancomycin-resistant strains [52]. These findings indicate the potential to reclaim already available antibiotics to fight AMR.

## 2. Polymeric Delivery Systems

Polymers are compounds with high molecular weight, generally composed of one or several structural units conjugated through covalent bonds. They easily form nanostructures when dispersed in water, primarily due to their hydrophobic/hydrophilic framework. Polymer-based nanostructures are usually highly biocompatible, stable in biological environments, and can merge a variety of functions in their structure (Figure 3) [53,54]. The polymeric systems applied to antibiotics provide an opportunity for developing nanobiotics with controlled drug release, improved pharmacokinetics, and enhanced antibacterial efficacy [55]. Antibiotics with poor water-solubility can be loaded in the core of the micelles or the membrane of polymeric vesicles, while hydrophilic antibiotics can be incorporated into the vesicles’ inner reservoir. Moreover, by engineering the core-forming segment of block copolymers, it is possible to load water-soluble agents, such as cationic antimicrobial peptides, into the core of micelles [56].

Polymeric delivery systems can be made of natural or synthetic polymers. Natural polymers are an attractive material for nanobiotics due to their availability, easy processability, and low price. However, the development of translatable formulations based on such materials may be limited due to potential impurities and batch-to-batch variations. Among natural polymers, the cationic polysaccharide, chitosan, has shown outstanding capacity for encapsulating antibiotics and promoting their efficacy [57,58,59]. Chitosan-based nanocarrier properties can be controlled by modifying side chain hydrophobicity to expose the positively charged moieties on the surface. These positively charged chitosan carriers can bind with bacteria, with a negatively charged membrane surface, to promote membrane disruption and intracellular delivery of drugs. Moreover, positively charged polymers can inhibit protein translation by interacting with mRNA and DNA inside the bacteria [60]. Thus, chitosan can contribute antimicrobial activity against both Gram-positive and Gram-negative bacteria [46,61], suggesting that antibiotic-loaded chitosan nanocarriers could have the potential to work synergistically to eliminate bacteria. In addition, due to the mucoadhesive properties of chitosan-coated delivery systems, drug retention time in mucosa sites increases. For example, daptomycin encapsulated into alginate nanoparticles coated with chitosan showed slower drug release, thus improving ocular delivery [62]. Similarly, polyelectrolyte complexes, formed by ionic interactions between polymers with opposite charges, were used for the delivery of nisin. The chitosan-pectin complex loaded with nisin blocked the growth of *S. aureus* for up to seven days in inhibition zone assays due to slow antibiotic release [63]. 

Chitosan has also been used in combination with nanoemulsion to incorporate *Thymus vulgaris* and *Syzgium aromaticum* essential oils [64]. The formulation was effective against MDR Gram-negative bacteria in vitro, such as methicillin-susceptible *S. aureus*, and MDR Gram-negative microorganisms, including carbapenem-resistant *Acinetobacter baumannii* and *Klebsiella pneumonia*. 

Synthetic polymers are a reliable and versatile alternative for developing nanobiotics. Nanoparticles based on polymers can be designed to incorporate antibiotics by controlling hydrophobic physical interactions [65], ion complexation [66], and π-π stacking [67] (Figure 4). Typical polymers used for physically loading antibiotics include poly(lactic-glycolic acid) (PLGA) for hydrophobic interactions [68]; polyethyleneimine, poly(styrene sulphonate) and poly(glutamic acid) for ion complexation [66,69,70]; and poly(benzyloxymethyl methyl glycolic acid) for π-π stacking [67]. Moreover, polymers with functional groups can be used for generating polymer-drug conjugates with tunable release rates. Particularly, the bonds between polymers and antibiotics can be designed to cleave upon sensing specific stimuli, such as acidic pH [71] or the presence of enzymes [72]. Such features are particularly useful for selectively activating antibiotic function in biofilms, which present low oxygen concentration, acidic microenvironments, and expression of specific enzymes [73,74]. Biocompatible polymers used for conjugating antibiotics include poly(hydroxyethyl methacrylate) [75], poly(amino acids) [76], poly(oxazolines) [77], and poly(urea) [78], among others. 

In a study polymeric micelles (<30 nm) composed of poly(ethylene glycol) methyl ether-block-poly(lactide-co-glycolide) (PLGA-PEG) were loaded with piperacillin/tazobactam [79]. The polymer-nanoparticles were more effective in inhibiting bacterial motility and had a lower minimum inhibitory concentration (MIC) value than free drugs. The nanoparticles were able to eradicate biofilm in seven bacterial strains at lower concentrations compared to the free drug. In another study, polyacrylic acid-based nanogels coated with cationic polyelectrolyte (bPEI) were loaded with tetracycline and lincomycin [80]. Owing to the enhanced accumulation on anionic bacteria surface, nanogels demonstrated enhanced antibacterial action and lower MIC in several species resistant to the free antibiotic.

Conventional polymeric systems may face the challenges of antibiotic leakage during circulation, insufficient release in the diseased site and off-target accumulation. Smart polymeric systems can be designed to store and release bactericidal agents in response to environmental triggers. The bacterial microenvironment provides unique intrinsic stimuli for such polymers, including acidic pH [71], high intracellular or extracellular glutathione (GSH) concentration [81], upregulated enzymes [82] and released toxins [83]. Polymers with acid-labile functional groups or linkers [84], reduction-sensitive di-sulfide bonds [85], or enzyme-cleavable linkages [86] could enhance the antimicrobial efficacy, improve antibiotics’ bioavailability and increase drug internalization and retention. For example, pH-responsive maleic anhydride (DA) modified poly(ethylene glycol)-block-polylysine was engineered to disassemble and undergo charge shifting at pH 6.0, releasing the azithromycin-conjugated poly(amidoamine) dendrimer (PAMAM) nanoparticles [84]. This system could enhance penetration and localization of antibiotics in biofilms and effectively eradicate bacterial count compared to free AZM. In another representative study, enzyme-degradable polymeric vesicles with β-lactamase (Bla)-cleavable side chain motifs in the hydrophobic block, selectively delivered vancomycin and inhibited the growth of MRSA strains [86].

Block copolymers [56], branched polymers [87], and dendrimers [88] can also facilitate the construction of nano-scaled carriers for delivering antibiotics. Particularly, amphiphilic block copolymers can be applied for self-assembling three-dimensional nanostructures in aqueous conditions, such as core-shell polymeric micelles and vesicles [89,90]. These nanostructures can improve the pharmacokinetics and the stability of the antibiotics in biological environments, reduce their toxicity, and enhance the selective delivery of antibiotics to bacteria. In addition, polymeric micelles usually offer a dense hydrophilic poly(ethylene glycol) (PEG)-shell, which can be a significant advantage for promoting access and penetration into biofilms to deliver antibiotics [74]. For instance, polyion complex micelles with anionic polyglutamic acid as the inner segment, substantially diminished the hemolytic action of cationic MSI-78 peptide against human red blood cells, and had comparable antibacterial activity to MSI-78 with little or no toxic effects [56].

## 3. Lipid-Based Delivery Systems

### 3.1. Phospholipid-Based Delivery Systems

Phospholipids are amphiphilic compounds that contain two fatty acid tails (hydrophobic portion) and a phosphate head (hydrophilic portion) linked by alcohol or glycerol molecules. Phospholipids self-assemble in lipid bilayers and are major constituents of cell membrane liposomes [91,92,93] (Figure 4). Due to their amphiphilic nature, liposomes can incorporate both hydrophilic and hydrophobic bioactive compounds. Hydrophilic compounds are entrapped within the internal aqueous core, whereas lipophilic compounds are incorporated into the lipid bilayers. Liposome size can vary from 10 nm to a few microns depending on composition, preparation method, and environmental conditions [94,95]. The most common methods used to fabricate liposomes include high-pressure homogenization, microfluidization, (a high shear mixer to blend oil, emulsifier, and water together) [96], ethanol injection, electro-spraying, and supercritical carbon dioxide [97]. 

Since liposomes can be produced with natural and safe materials, such as phospholipids from egg and soy, liposomes have been applied to encapsulate and deliver antimicrobial peptides [98,99,100,101,102] and antibiotic drugs [103,104]. Liu et al. incorporated antimicrobial peptide DP7-C and azithromycin in the lipid bilayer of liposomes [100]. The liposomes significantly reduced bacterial counts in an MRSA-infectious mouse model, possessed stronger antibacterial activity than DP7-C, and activated host immune responses against the bacteria. In addition, they did not exhibit obvious side effects. Co-encapsulation of two or more antibiotics in the same liposomal delivery vehicle can also enhance efficacy against MDR bacteria. For instance, a combination of colistin with ciprofloxacin was effective in tackling many of the colistin and ciprofloxacin-resistant bacterial pathogens [105].

However, liposomes are not stable against shear and external environmental stresses. They tend to break down in the physiological environment (in the presence of salts) and are highly sensitive to the pH change and presence of digestive enzymes experienced following GI administration [106,107]. In addition, liposomes adhere to each other, forming larger vesicles in suspension that result in leaked encapsulated components over time [108]. Their rapid clearance from the bloodstream by the reticuloendothelial system is another limitation. 

To overcome the above shortcomings, liposomal surfaces can be covered with biopolymer using the electrostatic layer-by-layer technique [99,109,110,111]. For example, liposomes covered with a cationic polymer (Eudragit E-100) layer-by-layer had a lower tendency to aggregate and were more stable [98]. Chitosan-coated liposomes containing dicloxacillin were evaluated for their efficacy against MRSA strains [112]. The high cell internalization of chitosan-coated liposomes enhanced dicloxacillin uptake and increased the efficacy of the antibiotic against MRSA. 

Besides biopolymers, hydrogel [113], polyethylene glycol [114,115], small molecule emulsifiers, and sterol [116,117] can enhance the stability and efficacy of liposomes. Cholesterol is frequently used to enhance stability during liposome formation, resulting in liposome membrane structural stability in the intestines [94,118]. In addition, in the absence of cholesterol, liposomes often interact with proteins, including albumin, transferrin, macroglobulin, and high-density lipoproteins. Such interactions destabilize liposome structure and, consequently, reduce their capacity as drug delivery systems [119]. 

Moreover, liposomes can also be modified as dual-coated food-grade formulations [99,119]. For example, whey protein and pectin-coated liposomes encapsulating antibacterial peptide (MccJ25) protected the peptide against gastrointestinal digestion and improved its controlled release [99]. Antimicrobial peptides can also be conjugated to liposomes. For example, a 24-residue peptide, named WLBU2, was conjugated to temoporfin-loaded liposomes [120]. It not only showed high antimicrobial activity but also high antimicrobial selectivity in the co-culture model. WLBU2-liposomes were efficient in temoporfin delivery due to their enhanced binding and fusion potential with bacteria. Liposomes have been evaluated for various applications, including oral, systemic, pulmonary and topical delivery, as well as treatment of intracellular infections. For example, efficacy of Arikayce (amikacin liposome inhalation suspension) against pulmonary nontuberculous mycobacteria infection was confirmed by clinical trial phase 3 [121]. As a result, liposomes are also extensively evaluated for the delivery of antibiotics including polymyxin [122], colistin [123,124,125], vancomycin [103], LL-37, indolicin [126] and polymyxin B [127]. Importantly, liposomes can deliver antibiotics into host cells to target intracellular infections. Li et al. showed that colistin-loaded liposomes were equally effective against *Pseudomonas aeruginosa* compared to a colistin solution in vitro. In vivo studies of *P. aeruginosa* tracheal-infection-bearing mice showed that treatment with colistin-loaded liposomes resulted in the survival of 50% of the mice up to 96 h post-infection, while none of the mice treated with empty liposomes or the colistin solution survived longer than 24 h post-infection. In addition, colistin-loaded liposomes reduced systemic exposure of the drug in mice, improving the safety of colistin [125].

Functionalized liposomes, using poly[(DEAEMA-co-BMA)-b-ManEMA], have been shown to improve the intracellular bacterial killing efficiency of streptomycin in the *Francisella* infected macrophage co-culture model compared to its free form [128]. Additionally, the decoration of 4-aminophenyl-alpha-d–manno-pyranoside (PAM) onto DOTAP liposome surface have also been demonstrated to prominently increase the cellular uptake of moxifloxacin hydrochloride into alveolar macrophages [129]. 

### 3.2. Emulsion-Based Delivery Systems

Emulsions can be defined as a system consisting of two immiscible liquids (commonly oil and water). The mixture of one liquid is dispersed as droplets in the other (Figure 5). These can be oil-in-water, water-in-oil [130], or double emulsions, such as water-in-oil-in-water [131]. A stable and heterogeneous dispersion of emulsion systems is formulated by the addition of emulsifiers. Emulsifiers are generally classed into two groups: macromolecule emulsifiers, e.g., proteins or polysaccharides [132,133], and small molecules, such as non-ionic surfactants [134]. 

Emulsion-based systems protect cargo against degradation, enhance water-solubility, and increase cell uptake [135,136,137]. In addition, they are useful for the encapsulation of lipophilic bioactive compounds, as these compounds can be easily dissolved within the hydrophobic core of lipid droplets. 

Nanoemulsions have received extensive interest as colloidal delivery systems due to the ease of scaling up and processing operations [138,139]. They are emulsion systems with droplet sizes on the nanometric scale, between 50 to 1000 nm [137,138,139]. Nanoemulsions, particularly oil-in-water, can encapsulate antimicrobial agents within the hydrophobic core of the lipid droplets (Figure 5) [138]. The choice of emulsifier significantly influences properties of emulsion, such as the ability of emulsion to promote dissolution of antimicrobials into the aqueous phase, which has been correlated with increased in vitro bactericidal activity [140]. The potential effects of nanoemulsions as delivery systems for antimicrobial agents, such as nisin [141] and peptides [131], in terms of enhancing their bioaccessibility, bioavailability, and/or bioactivity in vitro, have been explored. Achieving a particle size smaller than cell size (~500 nm) provides higher absorption of active compounds and higher particle uptake as a result of increased passive transport through intestinal epithelia. For example, double-layer emulsions containing whey protein and thyme essential oil droplets, as antimicrobial agents, were coated via electrostatic deposition of cationic chitosan around the anionic protein-coated droplets [142]. The emulsions exhibited prolonged antibacterial activity against two model food pathogens: *E. coli* and *S. aureus*. The total number of bacteria colonies after treatment with pure thyme oil for 24 h was 8.8 log CFU/mL, whereas the total number of *E. coli* colonies after treatment with double-layer emulsion was 1.2 log CFU/mL after only 5 h of treatment. A series of nanoemulsions containing either cationic, non-ionic surfactants or their blended components were formulated together with ethylenediaminetetraacetic acid in the aqueous phase. The combination of cationic/non-ionic surfactants provided the greatest nanoemulsion stability. This formulation (named NB-201) was more effective against *S. aureus* compared to a commercially available silver sulfadiazine solution in the infected wound mouse model [143]. Additionally, the topical treatment of MRSA infected wounds with NB-201 decreased the bacterial load both in vitro and in vivo in either murine or porcine models. NB-201 was able to significantly reduce the inflammatory cytokine secretions induced by MRSA infection [144]. Moreover, the study of Cecchini et al. demonstrated that the formulation of nanoemulsion based on *Minthostachys verticillata* (Griseb.) essential oil inhibited the growth of *S. aureus* without causing any toxicity to porcine and equine red blood cells [145]. 

### 3.3. Solid Lipid Nanoparticles and Nanostructure Lipid Carrier

Solid lipid nanoparticles (SLNs) and nanostructured lipid carriers (NLCs) refer to oil-in-water emulsions with a fully or partially solidified internal lipid phase, but dispersed aqueous phase [146]. SLNs are built from solid lipids, such as fatty acids, paraffin, glycerides, triglycerides, and waxes [147], whereas NLCs comprise a mixture of both solid and liquid lipids. The two main groups of liquid lipids used to formulate NLCs are saturated oils (such as medium-chain triglycerides, isopropyl palmitate, paraffin oil, and capric/caprylic triglycerides) and unsaturated oils (such as oleic acid and vegetable and seed oil) [148]. SLNs were developed to overcome the limitations of nanoemulsions (liquid oil droplets) and offer better protection for sensitive lipophilic bioactive compounds against degradation by both external stresses and gastrointestinal conditions [149]. Because of their crystalline structure, SLNs reduce the mobility and diffusion of bioactive components, thereby slowing down the release of entrapped ingredients and limiting the infiltration of pro-oxidants and other molecules that induce chemical degradation in the internal lipid core [150]. Furthermore, lipid crystallization retards the action of lipase on lipolysis, preventing burst release and extending the gastric retention time of the cargo [151,152].

Ban et al. fabricated SLNs using tristearin for the lipid phase and polyethylene glycol (PEG) as an emulsifier [153]. Curcumin-loaded long PEG-SLNs easily permeated mucus, resulting in a 12-fold increase in curcumin bioavailability in comparison to curcumin solution in rats. Wang et al. developed solid lipid-polymer hybrid nanoparticles (SLPN) as a delivery system for lipophilic bioactive compounds [154]. The SLPN internal lipid core was formed by either glyceride of fatty acids; bovine serum albumin-dextran Maillard conjugate was used as an emulsifier; and pectin as the secondary biopolymeric coating. The SLPNs formed by glycerides had high encapsulation efficiency and better-controlled release of curcumin than SLPNs formed by fatty acids. Rawat el al. also reported an increase in encapsulation efficiency of repaglinide, as multiple solid lipids were used to prepare the lipid nanoparticles [155]. Moreover, the SLPNs significantly improved the cellular uptake of curcumin without triggering any cytotoxicity.

Many biodegradable SLNs have been developed for loading hydrophilic antibacterial agents, including antibiotics, antimicrobial peptides, and herbals. For instance, SLN has been employed to deliver two different drugs, simultaneously, to the site of skin wound lesions. LL37, which is an antimicrobial peptide, was co-encapsulated with Serpin A1 into SLN using the emulsion solvent diffusion method; this combined formula displayed a synergistic effect against *S. aureus* and *E. coli* compared to the single-agent treatments [156]. Polymyxin B sulphate PLX-loaded SLN against *Pseudomonas aeruginosa* was formulated using a double emulsion method (o/w/o), with more than 90% loading efficiency [157]. The combination of lipids in the SLN was the main factor in modulating the occlusion effect and anti-microbial activities of PLX. The growth of some strains of *Pseudomonas aeruginosa* have been inhibited totally for PLX-SLN; however, free PLX showed a partial inhibition. 

Hosseini et al. synthesized SLN with encapsulated doxycycline (DOX-SLN) by double emulsion with palm wax and poloxamer as a surfactant [158]. The slow and continuous release of DOX from the SLNs was observed without any cytotoxicity. The formulation was effective against the diseases caused by intracellular bacteria, such as *B. melitensis*, in acute and chronic Wistar rat models.

Timicosin-loaded SLNs were more effective against *Streptococcus agalactiae* in the treatment of dairy cow mastitis than the drug alone [159]. Additionally, modification of the surface of SLNs encapsulating tilmicosin (TIMS-SLNs) using cationic surfactant dimethyl dioctadecyl ammonium chloride was shown to improve drug entrapment efficiency as well as enhance the controlled release property of the particles. The cationic TIMS-SLNs were formulated using a hot homogenization and ultrasonication technique with castor oil as a lipid matrix and polyvinyl alcohol as a surfactant. In vitro and in vivo antibacterial activities against avian *Pasteurella* were enhanced significantly in the modified TIMS-SLNs formulation compared to the unmodified one [160]. 

Still, SLNs have limitations, including limited drug-loading efficiency and non-uniform active diffusion from the particles [161]. To overcome these limitations, NLCs consisting of both solid and liquid lipids as the inner core at room temperature have been developed. A partial crystal NLC structure facilitates more space to accommodate bioactive ingredients, leading to less expulsion, higher loading capacity, and controlled release of the encapsulated compounds [162]. The lipid barriers around the incorporated component protected the loaded material from degradation. In comparison with other lipid-based nanocarriers, the higher density of solid NLC lipids enhanced the particles’ physicochemical stability and controlled release of loaded functional hydrophobic antimicrobial agents. Therefore, NLCs with high encapsulation efficiency provide a new avenue to enhance antimicrobial activity of hydrophobic antimicrobial agents. 

Sodium colistimethate and amikacin are effective antibiotics against a range of different resistant strains, such as *P. aeruginosa*, *K. pneumoniae*, and *A. baumannii*; however, their toxicity severely limits their use [163]. Encapsulation of these antibiotics into NLCs with sustained release of the active ingredient not only limited early drug degradation, but also led to lower plasmatic concentration of the drug, contributing to a safer toxicological profile. In an in vivo study, 1.5 × 10^8^ CFU/mL of drug-resistant *A. baumannii* clinical strain were injected intravenously and intraperitoneally into 48 mice. After 6 h, the bacteria were recovered from the blood and spleen. Sodium colistimethate-NLC lowered the bacterial load to the same level as a 10-fold higher dose of free sodium colistimethate. 

Cortesi et al. used NLC as a carrier of natural molecules with antimicrobial activity, including plumbagin, hydroquinon, eugenol, alpha-asarone, and alpha-tocopherol, for phytopathogenic control [164]. NLCs with bioactive agents were produced by the melt and ultrasonication method with the lipid mixtures, tristearin/miglyol, poloxamer 188 solution and polysorbate 80, as surfactants. The resulting NLCs containing plumbagin were safe and able to inhibit the growth of *F. oxysporum* significantly compared to unloaded agent. 

NLCs have been employed to encapsulate LL37 using the melt-emulsification method to improve effectiveness [91]. The nanoformulation improved the efficacy of LL37 in chronic wound treatment and enhanced its activity against *E. coli* in the infected wound area. Thus, both SLNs and NLCs are suitable carriers for topical delivery. NLCs coated with chitosan and sodium alginate have also been used to deliver tea tree oil as an antibacterial agent, mainly to avoid its evaporation from the formulation during storage [165].

Co-delivery of antioxidants with antibacterial agents in nanoparticles is a promising strategy for promoting wound healing, while reducing the development of resistance. For example, the antioxidant and antimicrobial activity of turmeric extract-loaded NLCs (particle size = 140 nm, encapsulation efficacy = 98%), compared to free turmeric extract, was tested [166]. In addition, curcumin encapsulated in nanoparticles exhibited higher antioxidant and radical scavenging activities than free curcumin solubilized in ethanol. Nanoparticles were also more active against the Gram-negative bacteria, *E. coli*, than free turmeric extract [167]. Recently, the hybrid nanoparticles with anti-microbial agent coating have been used as a novel approach for bacterial infection treatment [168]. It has been proposed that lipid-coated hybrid nanoparticles (LCHNPs) are the next-generation core–shell structured nanocarriers for the complex bacterial infection treatments [122]. The study of Bose et al. reported the use of lipid polymer hybrid nanoparticles decorating with cationic or zwitterionic lipids for doxycycline or vancomycin encapsulation. They observed that these modifications were able to enhance the bacterial killing efficiency of both antibiotic agents against *Mycobacterium smegmatis* or *Staphylococcus aureus* infected macrophages [169]. Another study by Tan et al. demonstrated that the loading of ampicillin into lipid–polymer hybrid nanoparticles (LPNs), with a unique combination of DOTAP lipid shell and PLGA polymeric core, significantly enhanced antibacterial activity of AMP against *Enterococcus faecalis* (*E. faecalis*) biofilms [170]. The implication of macrophage membranes as a coating or encapsulating material has also been demonstrated. The study by Li et. al. revealed that the coating of antimicrobial nanoparticles with macrophage membrane was able to increase antibacterial efficiency of triclosan and ciprofloxacin against *staphylococci* significantly by enhancing cellular uptake via Toll-like receptors [171]. This research proposed a new strategy using the cell membrane of immune cells as a coating or encapsulating materials as a new therapeutic window for preventing the development of bacterial drug resistance.

Overall, developments in lipid-based delivery systems provide a platform in nanoantibiotics with enhanced performance in terms of high drug (both hydrophilic and lipophilic) loading, as well as prolonged activity.

## 4. Preserving Antibiotic Potency by Complexing with Metals

The mode of action of classic antibiotics relies on specific bacterial processes, such as replication, transcription, and translation. However, the increase in microbial resistance to antibiotics necessitates urgent development of complex antimicrobial mechanisms to combat bacteria through multiple approaches. To achieve this, metals have been introduced as alternatives or additions to enhance the antibacterial activity of existing antibiotics and reactivate their antibiotic potency. 

Improved antibacterial activity of metal ions or metal nanoparticles can be achieved through three mechanisms. (1) Metal ions or metal nanoparticles interact directly with bacteria via various modes of action including membrane disruption (initiated by electrostatic interaction of positively charged silver ions with the negatively charged bacterial cell membrane), internal biomolecule damage, and the formation of reactive oxygen species [172]. (2) Metal-based nanoparticles can improve the physicochemical and biological characteristics of antibiotics complexed with them, including lipophilicity, stability, and half-life, to increase antibacterial activity [173]. (3) Metal ions or metal nanoparticles can inhibit the biofilm responsible for antibiotic resistance [174]. In this part of the review, the enhanced activity of antibiotics complexed with selected metal nanoparticles is discussed (Table 1), with a focus on silver and zinc metals, as the most reported antimicrobial agents in the literature.

### 4.1. Silver

Silver is not recognized as a trace metal in the human body and has no direct biological importance for general human health. However, based on its capacity to eliminate microbes by inhibiting their crucial life processes, silver has been widely used in the food industry and in healthcare since ancient times. Several silver salts, nanoparticles, and complexes have been approved as medicines to treat human infections. For example, silver sulfadiazine was the first FDA-approved silver-containing compound, used for topical treatment of burns to prevent infections. The mechanism of antibacterial activity of silver ions is still poorly understood. Generally, the positive charge of a silver cation can interact with the overall negatively charged bacterial cell wall and block cellular nutrient transport, leading to cell wall disruption [193]. When silver ions break through the cell wall, they can interfere with the cell’s respiratory system to destroy energy production [194]. Silver can also interact with DNA and inhibit cell division to stop cell replication [195]. Furthermore, silver ions have been reported to have a synergetic effect with antibiotics for enhanced antibacterial activity through multiple pathways [196,197,198,199,200,201]. The synergetic effect of silver ions has been evaluated with various antibiotics, including aminoglycosides, quinolones, and ß-lactams against *E. coli* (Table 2). The result showed enhancement of at least 40-fold with silver ions complexed with aminoglycosides, especially gentamicin and tobramycin, which indicates the efficiency of silver ions as an adjuvant. To better understand the antibacterial mechanism of silver ions on a molecular level, Wang et al. separated and identified the silver-binding proteins in *S. aureus* and found several key enzymes, including 6-phosphogluconate dehydrogenase and glycolysis, which provided insight into the sustainable susceptibility of bacteria to silver.

Silver nanoparticles comprise a reservoir of silver ions that can be slowly released to avoid rapid clearance; these could be applied in drug delivery systems to improve the stability of antibiotics. Based on their size and shape, silver nanoparticles can exhibit different antibacterial activity against Gram-positive and Gram-negative bacteria. Martinez-Gutierrez et al. reported spherical silver nanoparticles (20–25 nm) tested against Gram-positive bacteria, *Bacillus subtilis*, *Mycobacterium bovis*, *Mycobacterium smegmatis*, MRSA, *S. aureus*, and Gram-negative bacteria, *Acinetobacter baumannii*, *E. coli*, and *Pseudomonas aeruginosa* [202]. The nanoparticles exhibited better antibacterial activity against *B. subtilis, M. bovis*, MRSA and *P. aeruginosa* compared with antibiotics alone (Table 2).

To test the synergetic effect of silver nanoparticles in combination with antibiotics, Hassan et al. reported the combination of silver nanoparticles with cefotaxime to treat *E. coli*, *P. aeruginosa*, *S. aureus,* and *Staphylococcus arlettae* (Table 2) [175]. The combination exhibited better antibacterial activity compared with silver nanoparticles or cefotaxime alone. *P. aeruginosa* was sensitized by silver nanoparticles, leading to the increased activity of the combination with cefotaxime. For resistant *S. arlettae*, the disk diffusion test was applied with zone of inhibition, ZOI, values. The combination of silver nanoparticles and cefotaxime increased inhibition of 85% (ZOI 12 mm) compared with the cefotaxime alone (ZOI 7 mm), indicating the efficiency of silver nanoparticles as a synergetic agent. 

Kora et al. reported the combination of three different silver nanoparticles, capped by citrate, sodium dodecyl sulphate (SDS), and polyvinylpyrrolidone (PVP), respectively, mixed with antibiotics by loading discs containing streptomycin, ampicillin or tetracycline with nanosilver tested in discs diffusion against *E. coli* and *S. aureus* [176]. For Gram-negative *E. coli*, ampicillin activity increased 50% in combination with PVP-capped silver nanoparticles, followed by a 25% improvement of tetracycline with SDS-capped silver nanoparticles. For Gram-positive *S. aureus*, the highest percentage of enhancement was found with PVP-capped silver nanoparticles with streptomycin (45.4%), followed by tetracycline (23.8%) and ampicillin (13.3%). Further, Brown et al. reported that after the attachment of ampicillin to silver nanoparticle through the thioether moiety, the complex exhibited antibacterial activity against ampicillin-resistant *E. coli* and *P. aeruginosa* with an MBC of 1 µg/mL [177], which indicates the feasibility of silver nanoparticles to combat antibiotic resistance.

Kaur et al. reported synergistic bactericidal activity for vancomycin and amikacin silver nanoparticles [178]. Interestingly, vancomycin has no antibacterial activity against *E. coli*, as it is Gram-negative. Yet, in combination with PVP-capped silver nanoparticles, it inhibited the growth of bacteria to some extent, which indicated that silver nanoparticles can change the permeability of the cell membrane to allow the delivery of vancomycin into the cell. Moreover, Brasil et al. reported synergistic antibacterial activity of silver nanoparticles for azithromycin, levofloxacin, and tetracycline, with reductions in MIC of 37–97% against *S. aureus* [179]. Jamaran et al. reported the application of silver nanoparticles in treating cow’s mastitis. Twenty *S. aureus* strains were isolated from 46 milk samples and then used in discs diffusion test. Most of the strains were resistant to gentamicin and neomycin, but the combination of antibiotics with silver nanoparticles reduced the resistance of *S. aureus* by 15% and 45%, respectively [180].

Therefore, considering silver’s multi-target mode of action, it has great potential to combat antibacterial resistance and resensitize MRSA to antibiotics.

### 4.2. Zinc

Zinc is the only metal existing in all metalloenzyme classes, both in the human body and in microbes. Zinc can directly interact with the microbial membrane, which causes membrane destabilization and enhanced permeability, and with nucleic acid to cause enzyme deactivation in the bacterial respiratory system [203]. Bunnell et al. reported that zinc can block the mutator response to fight against antibiotic resistance [204]. The activity was tested in *E. coli* and *Klebsiella pneumoniae*, where zinc prevented the “save our ship” (SOS) response via inhibition of RecA. This prevented the repair of damaged DNA and led to the blockage of SOS-induced antibiotic resistance. Moreover, Vos et al. analyzed the effect of zinc in ciprofloxacin resistance, using ciprofloxacin-susceptible and ciprofloxacin-resistant *E coli* strains [205]. The presence of zinc improved the minimal selective concentration of ciprofloxacin resistance up to 5-fold. Although detail of the mechanism is poorly understood, ongoing research into zinc’s ability to fight against antibiotic resistance is warranted. Zinc also exhibited synergistic activity (decreased MIC by at least 2-fold) with ciprofloxacin, levofloxacin, moxifloxacin, norfloxacin, and ertapenem against *P. aeruginosa*—the pathogen responsible for urinary tract infections [206]. Levofloxacin, in particular, was improved; the MIC decreased 16-fold in the presence of 2.5 mM zinc sulphate.

In terms of zinc nanoparticles, Murugesan et al. reported the application of a zinc oxide nanocarrier for ciprofloxacin [181]. These nanoparticles proved to be a pH-sensitive drug delivery vehicle for controlled release of antibiotics for as long as 24 h. The ciprofloxacin-encapsulated nanoparticles also exhibited improved antibacterial activity compared to single ciprofloxacin or nanoparticles against *Bacillus cereus* and *P. aeruginosa*. The antibacterial activity of ciprofloxacin-encapsulated zinc nanoparticles was observed following treatment of *S. aureus* and *E. coli*: ZOI was increased 27% and 22%, respectively [182].

Another zinc oxide nanocarrier was reported by Mou et al., which exhibited antibacterial activity when the concentration was over 0.2 µg/mL [183]. The nanoparticles were pH-sensitive, and the slow-release time was more than 72 h. The cytotoxicity result for cell survival rate was more than 85% after 24 h of treatment with 0.8 mg/L zinc oxide nanoparticles, which indicated the efficiency and security of this nanoparticle. Thati et al. analyzed the antibacterial activity (via ZOI of *S. aureus*) of a wide range of antibiotics combined with zinc nanoparticles [184]. The results showed that β-lactam antibiotic activity improved by more than an 8 mm increase in ZOI, while penicillin G and amikacin had a 10 mm increase.

Zinc ions provide synergistic antibiotic and antibiofilm activity. Zinc nanoparticles exhibit great potential as nanocarriers to achieve slow and controlled release to increase the antibacterial activity of antibiotics.

### 4.3. Other Metal-Based Nanoparticles

Many other metal-based nanoparticles have been reported, with promising antibacterial activity. Only a few are discussed here, focusing on those that have been used to improve delivery of antibiotics.

Copper is a cofactor in many enzymes and plays a key role in defending oxidative damages in superoxide dismutase. Assadi et al. reported copper oxide nanoparticles as a drug delivery model [185]. Tetracycline was loaded into the model, which released gradually over five days through dialysis. Tetracycline-loaded copper oxide nanoparticles also exhibited better antibacterial activity compared with tetracycline alone against Gram-positive *S. aureus*.

Iron is the most abundant metal in the human body, involved in the transfer of oxygen from the lungs to tissues. Armijo et al. reported enhanced biofilm inhibitory activity against *P. aeruginosa* by loading tobramycin into alginate-capped iron oxide nanoparticles [174]. A study by Caamaño et al., with activity tested against *Streptococcus pneumoniae*, showed that the MIC of erythromycin decreased from 0.25 µg/mL to 0.12 µg/mL when combined with polyethylene glycol iron oxide nanoparticles (PEG-FeNPs), while PEG-FeNPs exhibited no inhibition [186]. Moreover, Wang et al. reported that a chitosan-PEG FeNP delivery system exhibited excellent penetrability into *S. aureus* biofilm due to the superparamagnetic performance of chitosan-PEG FeNPs [187].

Gold has been used in medicine, with reported antibacterial, anticancer, and anti-HIV therapeutic effects [207,208,209,210]. Due to the inherent features of gold nanoparticles, they can escape from bacterial efflux pumps [211]. Thus, gold nanoparticles can be a promising delivery system to enhance the antibacterial activity of antibiotics. Yang et al. reported broad-spectrum antibacterial effects from polycobaltocenium homopolymer-coated penicillin-G combined with gold nanoparticles (Table 3) [188]. Chamundeeswari et al. reported the application of gold nanoparticles with ampicillin, which increased efficacy and reduced ampicillin dosage by nearly 50% in vitro (MIC method), thus decreasing ampicillin’s side effects [189]. Further, gold nanoparticles have been reported to reactivate ampicillin to ampicillin-resistant *E. coli*. The MBC of gold nanoparticles functionalized with ampicillin (surface attached through thioether moiety) was 1 µg/mL, while gold nanoparticles alone had no antibacterial activity [177]. Moreover, gold nanoparticles have been conjugated with antibiotics through particles seeding growth in the presence of antibiotics, including streptomycin, kanamycin, gentamicin, vancomycin, and cefaclor. The MIC decreased by at least 30% for kanamycin against *E. coli DH5a*, *Micrococcus luteus*, and *S. aureus* [190], while the MIC and MBC of gentamicin decreased by 50% against *S. aureus* [212]. Cefaclor-gold nanoparticles exhibited MICs of 10 µg/mL and 100 µg/mL for *S. aureus* and *E. coli*, respectively, which were 5-fold decreases compared with cefaclor alone [191]. A significant improvement was observed for vancomycin, as well, as the MIC decreased 6.25-fold against vancomycin-resistant *S. aureus* [192].

To summarize, silver and other metal ions and their nanoparticles have exhibited great potential to improve the antibacterial activity of antibiotics [213,214,215,216]. As metals have multiple mechanisms against bacteria that are quite different from traditional antibiotics, they may revive ineffective antibiotics. Furthermore, as a result of their various antibacterial mechanisms, bacteria will need more specific mutations to escape the damage that metals can cause, reducing the potential for resistance.

## 5. Other Enhancers and Delivery Systems

In addition to natural and non-natural charged polymers, proteins can be employed in nanoparticles based on electrostatic interactions. Proteins can also form complexes between themselves or with other polymers through hydrogen bonding and hydrophobic interactions. Protein-based particles are biodegradable, relatively stable, non-toxic, and their surface can be easily modified. For example, curcumin has been encapsulated into spherical particles formed from collagen and gelatin. Encapsulation of curcumin improved its antioxidant and antimicrobial properties, broadening its potential against Gram-positive bacteria and fungi [217]. Mesoporous silica SBA-15 and recombinant human bone morphogenetic protein-2 (rhBMP-2) were loaded onto a zein-based scaffold [218]. In addition, hydroxypropyltrimethyl ammonium chloride derivatized chitosan (HACC) was incorporated into the nanoparticles as an anti-infective agent. The zein-HACC-S20 particles displayed effective antibacterial activity against *E. coli* and *S. aureus* lasting for at least 5 days; they were also able to slowly release rhBMP-2 for up to 27 days. Lysozyme, an antimicrobial agent against Gram-positive bacteria, was encapsulated into protein-based nanoparticles, gliadin cross-linked with cinnamaldehyde [219]. The nanoparticles enhanced the antimicrobial activity of cinnamaldehyde against *Listeria innocua*. Another protein-based delivery platform contained cross-linked gelatin nanoparticles as the core and a uniform red blood cell membrane as the shell [220]. Once vancomycin was loaded on the nanostructures, it was able to efficiently kill Gram-positive and gelatinase-positive pathogenic bacteria at small doses and with minimal toxicity.

AMPs are potent antimicrobials; however, most AMPs suffer from poor selectivity and severe cytotoxicity against mammalian cells. To overcome these issues, they can be modified with hydrophobic moieties, such as lipid (e.g., cholesterol), or hydrophobic amino acid residues (e.g., polyLeu) from amphiphiles, which can then self-assemble into nanoparticles [221]. Lei et al. conjugated myristic acid to the human AMP, human α-defensin 5, to impart its hydrophobicity using Gly or Lys linkers. The produced conjugate self-assembled into nanoparticles (56–80 nm). At a concentration of 100 µg/mL after 2 h, the self-assembled nanoparticles disrupted the bacterial membrane and cell structure, facilitating antibiotic entry, as evidenced by scanning electron microscopy [222]. The protein-based delivery system was also combined with superparamagnetic nanoparticles. Thus, vancomycin was conjugated to albumin-coated nanoparticles. The nanoparticles greatly improved the antimicrobial efficacy of vancomycin against vancomycin-resistant strains (MIC was reduced 10–100-fold). The magnetic-NP-conjugated vancomycin showed higher surface densities of antibiotics resulting in enhanced affinity towards bacteria, which allowed rapid permeabilization through the bacterial cell wall within 2 h [52]. No such effect was observed with free vancomycin.

Peptides, which are very small fragments of proteins, can also be used for antibiotic delivery. For example, supermolecule nanofibers can be formed based on the self-assembly of β-sheet-forming peptides via strong hydrophobic and π–π interactions. Chen et al. conjugated melittin, as a natural AMP, to poly(hydrophobic amino acids), including Leu, Lys, and Glu. The peptides adopted a β-sheet secondary structure, which increased their ability to permeate bacterial membranes, while reducing general toxicity. Unfortunately, the peptides were less effective in antimicrobial efficacy assays [221].

Co-delivering antibiotics active against Gram-positive bacteria with small molecule potentiators can improve the overall efficacy [223]. The potentiators, such as cyclic and linear peptides, disrupt the outer membrane of Gram-negative bacteria, allowing antibiotic entry and inhibiting bacterial growth. By binding to phosphatidylglycerol, they disrupt the outer and inner membrane, inhibiting essential enzymes and reducing biofilm formation. Furthermore, because of their selective binding to lipopolysaccharides, they display minimal cytotoxicity in clinical applications. Therefore, potentiators present a promising avenue for developing novel antimicrobial agents with efficacy against a broad range of Gram-negative, biofilm-forming, and MDR pathogens.

## 6. Conclusions

Due to the rapid emergence of AMR, the development of novel methods to improve the therapeutic efficacy of conventional antibiotics is of high importance. One of the most promising strategies is the delivery of antibiotics in specially designed delivery systems, often in the form of nano/microparticles. Delivery systems protect entrapped antibiotics from degradation and clearance, and deliver them directly to infected sites, tissues, or pathogens in a controlled manner without posing serious side effects to surrounding tissues. In addition, delivery systems can also act as activity enhancers and overcome bacterial resistance mechanisms, such as low permeability outer membranes, efflux mechanisms, biofilm formation, and inactivation of drugs. Nanosized delivery systems are also effective against intracellular pathogens.

A range of nanoengineered drug delivery systems have been examined to deliver antibiotics, including polymer-, lipid-, and metal-based systems. In most cases, delivery resulted in reduced toxicity and degradation of antibiotics; however, the antimicrobial efficacy of antibiotics was not always improved. Still, some delivery systems and antimicrobial activity enhancers improved the efficacy of antibiotics, as they displayed antimicrobial potency, for example, silver or chitosan. Antioxidants, or mixtures of antimicrobial agents, have been formulated together to further boost the antimicrobial activity of components and prevent drug resistance. Delivering multiple drugs in one carrier offers the opportunity to utilize multiple mechanisms of action against bacteria and, therefore, overcome MDR. In addition, combining different delivery strategies in one formulation, such as polymer-coated lipid-based nanoparticles, can deliver the advantages of each system, inducing stronger and safer antimicrobial action. However, the safety and toxicity, required dose, and effective routes of administration of the multiple combinations of antibiotics and delivery systems need to be evaluated more carefully. Furthermore, the effectiveness of these systems against bacteria, especially Gram-negative bacteria, is often insufficient and requires further development to reach clinical applications. In particular, testing for in vivo efficacy in rodent infection models is often lacking from publications that have reported promising new systems. It is essential to develop precisely delivered antibiotics in order to save millions of lives in the near future: a future that presents a rapidly growing number of life-threatening infections.

## Figures and Tables

**Figure 1 antibiotics-11-00412-f001:**
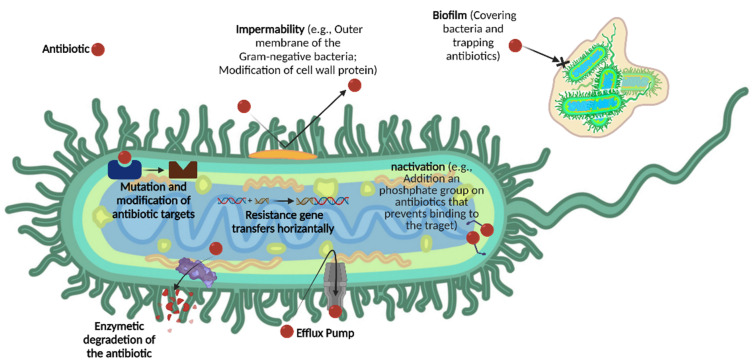
The seven most common mechanisms of antimicrobial resistance (AMR).

**Figure 2 antibiotics-11-00412-f002:**
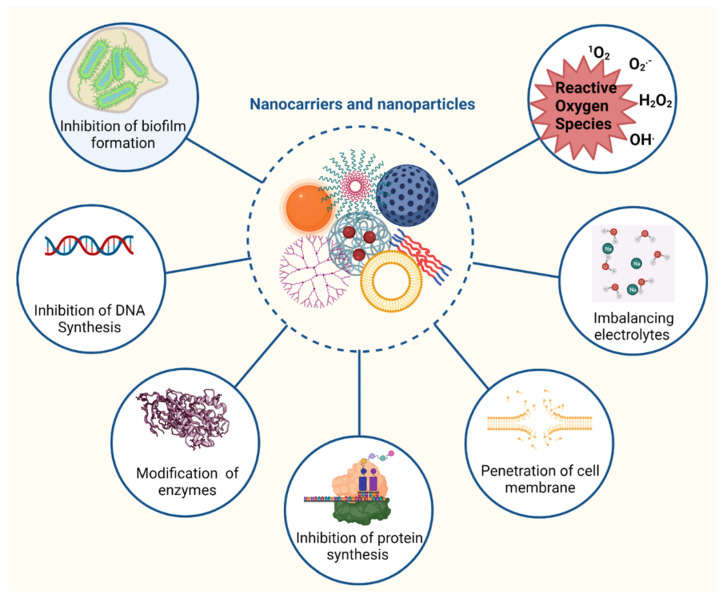
The most common mode of actions of antibiotics loaded in nanoparticles.

**Figure 3 antibiotics-11-00412-f003:**
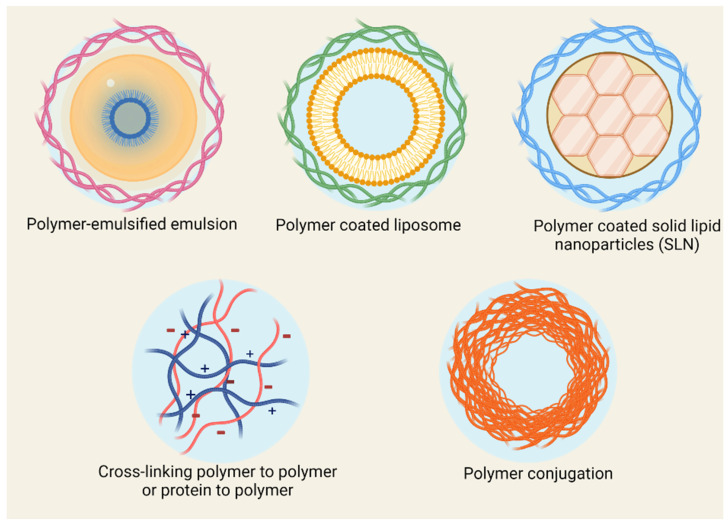
Polymer-based antibiotic delivery systems.

**Figure 4 antibiotics-11-00412-f004:**
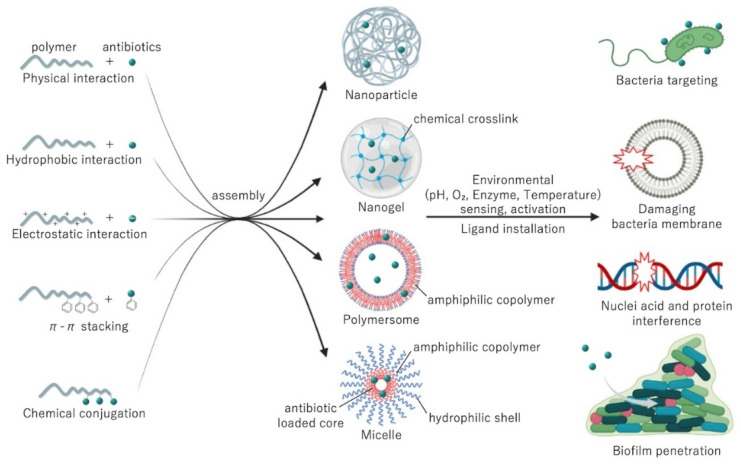
The assembly strategies and mechanism of actions of antibiotics loaded in polymeric delivery systems.

**Figure 5 antibiotics-11-00412-f005:**
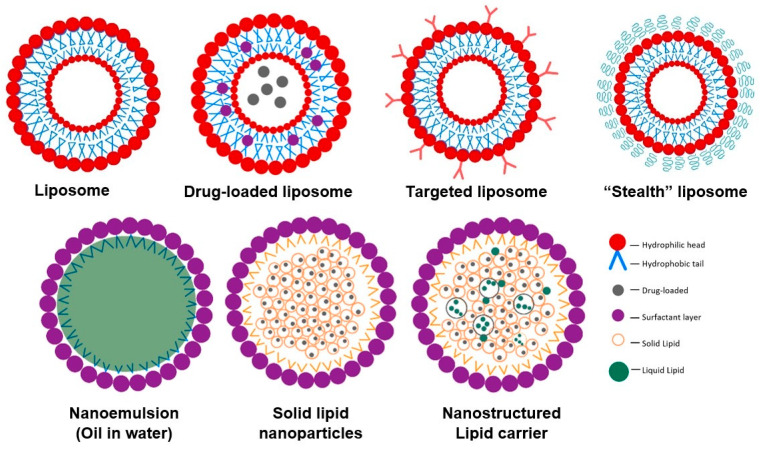
Schematic representation of antimicrobial agent lipid-based delivery systems.

**Table 1 antibiotics-11-00412-t001:** Antibacterial potency of metal nanoparticles complexed with antibiotics.

Nanoparticles	Antibiotic	Bacteria	Effect	References
Silver	Cefotaxime	*E. coli*, *P. aeruginosa*, *S. aureus*, *S. arlettae*	Synergistic effects: 13% for *S. aureus*; 17% for *E. coli*; 85% for *S. arlettae*	[175]
	Streptomycin Ampicillin Tetracycline	*E. coli*, *S. aureus*	Ampicillin improved up to 50% against *E. coli*, streptomycin improved 45% against *S. aureus*	[176]
	Ampicillin	Ampicillin-resistant *E. coli*, *P. aeruginosa*	MBC ^1^ 1 µg/mL	[177]
	Vancomycin Amikacin	*E. coli*, *S. aureus*	A promising carrier to deliver antibiotics into the bacteria cell	[178]
	Azithromycin Levofloxacin Tetracycline	*S. aureus*	Reduction in MIC ^2^ by 37–97%	[179]
	Gentamicin Neomycin	*S. aureus*	Synergistic effect; activity against antibiotic resistance	[180]
Zinc	Ciprofloxacin	*Bacillus cereus*, *P. aeruginosa*	Synergistic effect; pH-sensitive; slow-release nanocarrier	[181]
	Ciprofloxacin	*S. aureus*, *E. coli*	Synergistic effect: ZOI ^3^ increased 27% and 22% against *S. aureus* *E. coli*, respectively	[182]
	None	None	Improved Safety; pH-sensitive; slow-release nanocarrier	[183]
	β-lactams Cephalosporin’s Amino glycosides	*S. aureus*	Synergistic effect: ZOI increased by at least 7 mm	[184]
Copper	Tetracycline	*S. aureus*	Improved antibacterial activity; slow release nanocarrier	[185]
Iron	Tobramycin	*P. aeruginosa*	Enhanced biofilm inhibitory activity	[174]
	Erythromycin	*S. pneumoniae*	MIC decreased 50%	[186]
	None	*S. aureus*	Excellent penetrability into biofilm	[187]
Gold	Penicillin-G	*S. aureus*, *E. coli*, *K. pneumonia*, *P. vulgaris*	Significant improvement of antibacterial activity	[188]
	Ampicillin	*S. aureus*, *E. coli*, *K. mobilis*	50% reduction in dosage with the same antibacterial activity	[189]
	Ampicillin	Ampicillin-resistant *E. coli*	Reactivate ampicillin with MBC 1 µg/mL	[177]
	Kanamycin Gentamicin	*E. coli* DH5a, *Micrococcus luteus*, *S. aureus*	At least 30% decrease in MIC for kanamycin; 50% decrease in MIC and MBC for gentamicin against *S. aureus*	[190]
	Cefaclor	*S. aureus*, *E. coli*	5-fold decrease in MIC	[191]
	Vancomycin	Vancomycin-resistant *S. aureus*	6-fold decrease in MIC	[192]

^1^ MBC, minimum bactericidal concentration; ^2^ MIC, minimum inhibitory concentration; ^3^ ZOI, Zone of inhibition.

**Table 2 antibiotics-11-00412-t002:** Synergetic effects of silver ions complexed with antibiotics.

Bacteria	Antibiotics	Method	Antibacterial Activity		Reference
			Silver + Antibiotics	Antibiotics	
*E. coli*	Gentamicin	MIC (µg/mL)	0.07 ± 0.02	2.8 ± 0.3	[165]
	Tobramycin		0.08 ± 0.02	3.2 ± 0.5	
	Kanamycin		0.39 ± 0.10	6.8 ± 0.7	
	Streptomycin		0.64 ± 0.14	19 ± 4	
	Spectinomycin		12 ± 1	20 ± 0.0	
	Norfloxacin		72 ± 5	88± 9	
	Nalidixic Acid		3.1 ± 0.3	3.8 ± 0.5	
	Ampicillin		2.5 ± 0.4	3.2 ± 0.3	
	Chloramphenicol		3.5 ± 0.0	5.3 ± 0.4	
	Tetracycline		1.0 ± 0.0	1.8 ± 0.2	
	Amikacin		0.5	0.5	[172]
	Cefotaxime	ZOI (mm)	11.3 ± 2.1	10 ± 2	[144]
*S. aureus*	Gentamicin	MIC (µg/mL)	0.7	1	[172]
	Cefotaxime	ZOI (mm)	14 ± 1	12 ± 2	[144]
*P. aeruginosa*	Amikacin	MIC (µg/mL)	0.4	1	[172]
	Cefotaxime	ZOI (mm)	9.3 ± 0.6	0	[144]
*S. arlettae*	Cefotaxime	ZOI (mm)	12 ± 3	6.7 ± 0.6	[144]
*B. subtilis*	Gentamicin	MIC (µg/mL)	1.7	32	[172]
*M. bovis*	Rifampicin		1.1	0.5	
*M. smegmatis*	Rifampicin		0.5	0.85	
Methicillin-resistant *S. aureus*	Gentamicin		0.5	64	
*A. baumannii*	Amikacin		0.4	0.06	

**Table 3 antibiotics-11-00412-t003:** The increased antibacterial efficacy of penicillin with gold nanoparticles.

Compound	MIC (µg/mL)			
	*S. aureus*	*E. coli*	*K. pneumonia*	*P. vulgaris*
Penicillin-G	15.8	20.4	23.2	22.6
PCo-Penicillin ^1^	6.4	8.3	8.9	7.8
PCo-Penicillin + AuNPS	2.6	4.5	5.4	4.9

^1^ PCo-Penicillin, Polycobaltocenium homopolymer coated penicillin-G.

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
