# Peer review of "Antimicrobial Activity Enhancers: Towards Smart Delivery of Antimicrobial Agents"

_antibiotics, 2022, doi:10.3390/antibiotics11030412_

Round 1

Reviewer 1 Report

Dear Autors,

It was a pleasure to read the paper  entitled Antimicrobial Activity Enhancers: Towards Smart Delivery of Antimicrobial Agents.

There are many studies and reviews with topics on agents that increase the antimicrobial efficacy of antibiotics as presented here as bibliographic references. Of course they discuss the very important effect of nature, size, morphology obtained controlled by different methods of obtaining systems based on polymers, lipids and metals on animicrobial activity.

In this paper, I appreciate that the role in the intelligent delivery of antimicrobial agents and the mechanism of antimicrobial activity have been mainly discussed. In addition, delivery systems can also act as activity enhancers and overcome bacterial resistance mechanisms such as low permeability outer membranes, efflux mechanisms, biofilm formation, and drug inactivation.

Indeed, one of the most promising strategies is the delivery of antibiotics in specially designed delivery systems, often in the form of nano / microparticles. They protect antibiotics from degradation and disposal and deliver them targeted to infected areas, tissues or pathogens in a controlled manner without serious side effects on surrounding tissues.

In other words, I recommend the publication of this review, which contributes to the intelligent delivery of antimicrobial agents and antimicrobial resistance to antibiotics. The paper is well organized and comprehensively described and the references are current and appropriate with previous related papers.

Author Response

RESPONSE: Thank you for your favorable comments.

Reviewer 2 Report

Comments to the authors:

  1. please be consistent with the template formatting and follow the guidelines to authors
  2. Figure legends should be comprehensive and should be no need to go back to the manuscript to understand the figures. please revise accordingly
  3. lines 111-113 please revise for clarity and readability
  4. "The combination of nanoscale delivery system with antibiotics, known as “nanobiotics”, provides a high surface-to-volume ratio and the possibility of surface functionalization, contributing to high bacterial affinity and efficient antibacterial activity", I don't believe this is the only reason the use of nanoparticles for therapeutic delivery as a surface to volume ratio is one of many and mainly apply to catalyst
  5. Section 1.4 doesn't add any value to the other sections and it is repeated elsewhere in the review
  6. Figure 2 implies that nanomaterials target pathways. from the figure, it doesn't implicate this rather than it lists the targets of nanomaterials. please revise and revise the figure legend. 
  7. discussing ROS generated from nanomaterials. authors are advised to give specific examples and list how the generated ROSs affect different microbes, please add specific examples to the section
  8. Section 2 lists types of nanomaterials and it needs to be revised with specific examples otherwise it doesn't add any values to the review
  9. Explain the value of coatings as an antimicrobial on the lipid-based nanoparticles etc. what is the influence of nanoparticles coating on antimicrobial activities
  10. It is not clear to me the point of including the interactions of antibiotics with nanoparticles
  11. I don't get the idea of phospholipid based delivery system as a separate section rather than including lipid-based nanomaterials
  12. "microfluidization" please define new terms
  13. lines 258-259 please revise for clarity and readability
  14. It seems section 3 is very hard to follow is it about the formulation of nanomaterials rather than giving precise examples on how each nanoparticle was used for the delivery of antibiotics to specific species
  15. Lines 416-420 please revise for readability and clarity
  16. section 4.1 although authors are discussing silver ions, it is confusing whether it is silver ions or silver nanoparticles as the review discusses nanomaterials not ions
  17.  please be consistent with the language sometimes authors use British English mixed with American English
  18. section 4.2 similar story both ions and nanoparticles are part of the section
  19. Section 5 what does the author means by enhancers 
  20. The abstract needs significant changes as it doesn't represent the manuscript
  21. it is advised that authors should include a stimuli-responsive aspect of drug delivery 
  22. what is the interaction mode between nanoparticles and bacterial membrane
  23. There should be a section about antibacterial NPs against biofilms forming bacteria, intracellular bacteria etc   
  24. There are many recent reviews similar to this topic and I can't see the novelty and the new added value in this review (https://doi.org/10.3390/pharmaceutics13111795; https://doi.org/10.1080/14787210.2021.1908125; doi: 10.3390/nano10030560) and many more
  25.  

Author Response

Reviewer # 2

Comment # 1

please be consistent with the template formatting and follow the guidelines to authors.

RESPONSE: It was corrected as requested.

Comment # 2

Figure legends should be comprehensive and should be no need to go back to the manuscript to understand the figures. please revise accordingly.

RESPONSE: Figure legends were changed as follow:

a)

Figure 1. Mechanisms of antimicrobial resistance (AMR).

was changed to:

Figure 1. The seven most common mechanisms of antimicrobial resistance (AMR).

b)

Figure 2. Antimicrobial activity of nanobiotics.

was changed to:

Figure 2. The most common mode of actions of antibiotics loaded in nanoparticles.

c)

Figure 3. Polymer-based delivery systems.

was changed to:

Figure 3. Polymer-based antibiotic delivery systems.

d)

Figure 4. Interaction of antibiotics with natural and synthetic polymers.

was changed to:

Figure 4. The assembly strategies and mechanism of actions of antibiotics loaded in polymeric delivery systems.

Comment # 3

lines 111-113 please revise for clarity and readability.

RESPONSE: Lines 111-113 were revised as requested.

From:

“In contrast, physical entrapment of antimicrobial agents within the carrier protects their biological activity; but may allow premature drug release. Therefore, careful de-livery system optimization is required.”

Changed to:

“In contrast, physical entrapment of antimicrobial agents within the carrier protects their biological activity. However, the lack of chemical conjugation between antimicrobial agents and carriers may allow premature drug release. Therefore, careful optimization of a delivery system is required.”

Comment # 4

"The combination of the nanoscale delivery system with antibiotics, known as “nanobiotics”, provides a high surface-to-volume ratio and the possibility of surface functionalization, contributing to high bacterial affinity and efficient antibacterial activity", I don't believe this is the only reason the use of nanoparticles for therapeutic delivery as a surface to volume ratio is one of many and mainly apply to catalyst.

RESPONSE: Yes, we agree with the reviewer’s comment. To clarify this issue please see the following section in the manuscript:

“The combination of the nanoscale delivery system with antibiotics, known as “nanobiotics”, provides a high surface-to-volume ratio and the possibility of surface functionalization, contributing to high bacterial affinity and efficient antibacterial activity. The efficient transport of nanoparticles through the endothelium of the inflammatory site means that lower drug doses can be administered. The high permeability of the nanoscale delivery system through biological greatly improves antibiotics efficacy against intracellular pathogens. In addition, the surface of such delivery systems can be modified with ligands of tissue-specific receptors to allow encapsulated drugs to be transported to desired sites of action and improve cellular uptake.”

Comment # 5

Section 1.4 doesn't add any value to the other sections and it is repeated elsewhere in the review.

RESPONSE: Section 1.4 is describing the antimicrobial mechanisms of the nanoparticles and we believe it should be maintained in the manuscript. For example, it discusses “ROS generated from nanomaterials”, as requested in the reviewer comment 7.

Comment # 6

Figure 2 implies that nanomaterials target pathways. from the figure, it doesn't implicate this rather than it lists the targets of nanomaterials. please revise and revise the figure legend. 

RESPONSE: We have modified the figure legend accordingly (for details see responses to comment 2).

Comment # 7

discussing ROS generated from nanomaterials. authors are advised to give specific examples and list how the generated ROSs affect different microbes, please add specific examples to the section.

RESPONSE: To clarify this issue following section of the manuscript was modified:

From:

“The induction of oxidative stress by reactive oxygen species (ROS) is a key aspect of nanoparticle action against bacteria. The photocatalytic production of ROS damages vital components of the pathogen, including membranes, lipids, and proteins.”

Changed to:

“The induction of oxidative stress by reactive oxygen species (ROS) is a key aspect of nanoparticle action against bacteria. ROS such as superoxide anion, hydrogen peroxide, and hydroxyl radicles damages vital components of the pathogen, including membranes, lipids, and proteins. ROS inhibit the replication of DNA, amino acid synthesis, and damage microbial cell membrane [1]. For example, silver nanoparticles induced the formation of hydroxyl radicals in Gram-negative bacteria such as Pseudomonas aeruginosa and Shigella flexneri; and Gram-positive bacteria including Staphylococcus aureus and Streptococcus pneumoniae while iron oxide nanoparticles triggered the formation of hydroxyl radical and hydrogen peroxide against Bacillus subtilis and Escherichia coli [2,3]. ”

Comment # 8

Section 2 lists types of nanomaterials and it needs to be revised with specific examples otherwise it doesn't add any value to the review.

RESPONSE: The following section was added in line 239-260.

“In another study, polyacrylic acid-based nanogels coated with cationic polyelectrolyte (bPEI) were loaded with tetracycline and lincomycin [4]. Owing to the enhanced accumulation on anionic bacteria surface, nanogels demonstrated enhanced antibacterial action and lower MIC in several species resistant to the free antibiotic.

Conventional polymeric systems may face the challenges of antibiotic leakage during circulation, insufficient release in diseased site and off-target accumulation. Smart polymeric systems can be designed to store and release bactericidal agents in response to environmental triggers. The bacterial microenvironment provides unique intrinsic stimuli for such polymers, including acidic pH, high intracellular or extracellular glutathione (GSH) concentration [5], upregulated enzymes [6] and released toxins [7]. Polymers with acid-labile functional groups or linkers [8], reduction-sensitive di-sulfide bonds [9] or enzyme-cleavable linkages [10] could enhance the antimicrobial efficacy, improve antibiotics bioavailability and increase drug internalization and retention. For example, pH-responsive maleic anhydride (DA) modified poly(ethylene glycol)-block-polylysine was engineered to disassemble and undergo charge shifting at pH 6.0, releasing the azithromycin-conjugated poly(amidoamine) dendrimer (PAMAM) nanoparticles [8]. This system could enhance penetration and localization of antibiotics in biofilms and effectively eradicate bacterial count compared to free AZM. In another representative study,  enzyme-degradable polymeric vesicles with β-lactamase (Bla)-cleavable side chain motifs in the hydrophobic block, selectively delivered vancomycin and inhibited growth of MRSA strains [10].”

Comment # 9

Explain the value of coatings as an antimicrobial on the lipid-based nanoparticles etc. what is the influence of nanoparticles coating on antimicrobial activities.

RESPONSE: Thank you for your insightful suggestion. We have added the suggested information into the manuscript as followed:

“The functionalized liposome, using  poly[(DEAEMA-co-BMA)-b-ManEMA], have been shown to improve the intracellular bacterial killing efficiency of streptomycin in the Francisella infected macrophage co-culture model compared to its free form [11]. Additionally, the decoration of 4-aminophenyl-alpha-d–manno-pyranoside (PAM) onto DOTAP liposome surface have been also demonstrated to prominently increase the cellular uptake of moxifloxacin hydrochloride into alveolar macrophages [12].”

Also:

“Recently, the hybrid nanoparticles with anti-microbial agent coating have been used as a novel approach for bacterial infection treatment [13]. It has been proposed that lipid-coated hybrid nanoparticles (LCHNPs) are the next-generation core–shell structured nanocarriers for the complex bacterial infection treatments. The study of Bose et. al reported the use of lipid polymer hybrid nanoparticles decorating with cationic or zwitterionic lipids for doxycycline or vancomycin encapsulation. They observed that these modifications were able to enhance the bacterial killing efficiency of both antibiotic agents against Mycobacterium smegmatis or Staphylococcus aureus infected macrophages [14]. Other study by Tan et. al demonstrated that the loading of ampicillin into Lipid-Polymer hybrid Nanoparticles (LPNs), with a unique combination of DOTAP lipid shell and PLGA polymeric core, was significantly enhance anti-bacterial activity of AMP against Enterococcus faecalis (E. faecalis) biofilms [15]. The implication of macrophage membranes as a coating or encapsulating material has been also demonstrated. The study by Li et. al. revealed that the coating of antimicrobial nanoparticles with macrophage membrane was able to increase antibacterial efficiency of triclosan and ciprofloxacin against staphylococci significantly by enhancing cellular uptake via Toll-like receptor [16]. This research proposed the new strategy using cell membrane of immune cells as a coating or encapsulating materials as a new therapeutic window for preventing the development of bacterial drug resistance.”

Comment # 10

It is not clear to me the point of including the interactions of antibiotics with nanoparticles.

RESPONSE: These interactions are used to entrap antibiotic in polymer-based delivery systems, and therefore they are crucial for production, stability and functionalization of these delivery systems (please see Figure 4).

Comment # 11

I don't get the idea of a phospholipid-based delivery system as a separate section rather than including lipid-based nanomaterials.

RESPONSE: Liposomes (phospholipid-based delivery system) and lipid nanoparticles (LNPs) are similar by design, but different in composition and function. Unlike, other lipid nanocarriers, using surfactant as a particle stabilizer, liposome is a self-formed particle using the arrangement of phospholipids into lipid bilayer surrounding an aqueous pocket [17,18]. Thus, liposome can encapsulate both hydrophobic and hydrophilic drugs. On the other hand, some LNPs typically displays micelle-like structure, which solely encapsulate drug molecules in a non-aqueous phase [19]. Additionally, liposome is the first nanocarrier that have been approved by FDA for the use as a drug delivery system. Therefore, we would like to highlight these differences and the important of liposomes to other lipid carriers by separating this section as a Phospholipid-Based Delivery Systems.

Comment # 12

"microfluidization" please define new terms.

RESPONSE: Definition was added as requested:

“ Microfluidization” is a common high-energy technique used to produce micro and nanoscale size materials [20]. This approach involves the implication of a high shear mixer to form a coarse O/W emulsion by blending oil, emulsifier, and water together.

Comment # 13

lines 258-259 please revise for clarity and readability.

RESPONSE: Lines 258-259 were revised as requested.

Was:

“Liu et al. anchored antimicrobial peptide DP7-C and azithromycin in the lipid bilayer of liposomes.”

Changed to:

“Liu et al. incorporated antimicrobial peptide DP7-C and azithromycin in the lipid bilayer of liposomes.”

Comment # 14

It seems section 3 is very hard to follow is it about the formulation of nanomaterials rather than giving precise examples on how each nanoparticle was used for the delivery of antibiotics to specific species.

RESPONSE: Thank you for your constructive comment. We have added more examples into this section of the manuscript as follow:

“Liposomes have been evaluated for various applications, including oral, systemic, pulmonary and topical delivery, as well as treatment of intracellular infections. For example, efficacy of Arikayce (amikacin liposome inhalation suspension) against pulmonary nontuberculous mycobacteria infection was confirmed by clinical trial phase 3 [21]. As a result, liposomes are also extensively evaluated for the delivery of antibiotics including polymyxin [22], colistin [23-25], vancomycin [26], LL-37, indolicin [27] and polymyxin B [28]. Importantly, liposomes can deliver antibiotics into host cells to target intracellular infections. Li et al. showed that colistin-loaded liposomes were equally effective against Pseudomonas aeruginosa compared to a colistin solution in vitro. In vivo studies of P. aeruginosa tracheal-infection-bearing mice showed that treatment with colistin-loaded liposomes resulted in the survival of 50% of the mice up to 96 h post-infection, while none of the mice treated with empty liposomes or the colistin solution survived longer than 24 h post-infection. In addition, colistin-loaded liposomes reduced systemic exposure of the drug in mice, improving the safety of colistin [25].”

Also,

“A series of nanoemulsions containing either cationic, non-ionic surfactants or their blended components were formulated together with ethylenediaminetetraacetic acid in the aqueous phase. The combination of cationic/non-ionic surfactants provided the greatest nanoemulsion stability. This formulation (named NB-201) was more effective against S. aureus compared to a commercially available silver sulfadiazine solution in the infected wound mouse model [29]. Additionally, the topical treatment of MRSA infected wounds with NB-201 decreased the bacterial load both in vitro and in vivo in either murine or porcine models. NB-201 was able to significantly reduce the inflammatory cytokine secretions induced by MRSA infection [30]. Moreover, the study of Cecchini et. al demonstrated that the formulation of nanoemulsion based on Minthostachys verticillata (Griseb.) eessential oil inhibited the growth of S. aureus without causing any toxicity to porcine and equine red blood cells [31].”

Comment # 15

Lines 416-420 please revise for readability and clarity.

RESPONSE: The following sentence has been moved to the end of section 3 (the sentence was accidently shifted to the inappropriate section of the manuscript):

“Overall, developments in lipid-based delivery systems provide a platform in nanoantibiotics with enhanced performance in terms of high drug (both hydrophilic and lipophilic) loading, as well as prolonged activity.”

Comment # 16

section 4.1 although authors are discussing silver ions, it is confusing whether it is silver ions or silver nanoparticles as the review discusses nanomaterials, not ions.

RESPONSE: We thank the reviewer for the comment, however the review doesn’t focus only on nanomaterials but on “delivery systems or/and activity enhancers”, therefore we think it is an appropriate to mention both, metal nanoparticles and metal ions, especially as in aqueous condition metal ions are released from metal particles. 

Comment # 17

please be consistent with the language sometimes authors use British English mixed with American English.

RESPONSE: The American English has been selected for the consistency.

Comment # 18

Section 4.2 similar story both ions and nanoparticles are part of the section.

RESPONSE: Please see response to the comment 16.

Comment # 19

Section 5 what does the author means by enhancers.

RESPONSE: Enhancers are any molecules increasing activity of antibiotics, while delivery systems are limited to carrier-based systems. To clarify this issue following sentence was modified in the manuscript:

From:

“The development of new antibiotics with broad mechanisms of action or utilizing delivery systems to improve the efficacy of existing antimicrobial agents are the main approaches to fight AMR.”

Changed to:

“The development of new antibiotics with broad mechanisms of action or utilizing activity enhancers (e.g. silver ion), including delivery systems (e.g. liposomes), to improve the efficacy of existing antimicrobial agents are the main approaches to fight AMR.”

Comment # 20

The abstract needs significant changes as it doesn't represent the manuscript.

RESPONSE: The abstract was corrected as requested.

From:

“The development of effective treatments against infectious diseases is an extensive and ongoing process due to the rapid adaptation of bacteria to antibiotic-based therapies. However, appropriately designed antibiotic delivery systems or/and activity enhancers can overcome antimicrobial resistance and decrease the chance of contributing to further bacterial resistance. This review provides insights into various antibiotic activity/delivery enhancers, including polymer, lipid, silver, and other metals, designed to reduce the adverse effects of antibiotics and improve formulation stability and efficacy against multidrug-resistant bacteria.”

Changed to:

“The development of effective treatments against infectious diseases is an extensive and ongoing process due to the rapid adaptation of bacteria to antibiotic-based therapies. However, appropriately designed activity enhancers, including antibiotic delivery systems, can increase the effectiveness of current antibiotics, overcoming antimicrobial resistance and decreasing the chance of contributing to further bacterial resistance. The activity/delivery enhancers improve drug absorption, allow targeted antibiotic delivery, improve their tissue and biofilm penetration and reduce side effects. This review provides insights into various antibiotic activity enhancers, including polymer, lipid, and silver-based systems, designed to reduce the adverse effects of antibiotics and improve formulation stability and efficacy against multidrug-resistant bacteria.”

Comment # 21

it is advised that authors should include a stimuli-responsive aspect of drug delivery.

RESPONSE: The following section was added to clarify this issue.:

“Smart polymeric systems can be designed to store and release bactericidal agents in response to environmental triggers. The bacterial microenvironment provides unique intrinsic stimuli for such polymers, including acidic pH, high intracellular or extracellular glutathione (GSH) concentration [5], upregulated enzymes [6] and released toxins [7]. Polymers with acid-labile functional groups or linkers [8], reduction-sensitive di-sulfide bonds [9] or enzyme-cleavable linkages [10] could enhance the antimicrobial efficacy, improve antibiotics bioavailability and increase drug internalization and retention. For example, pH-responsive maleic anhydride (DA) modified poly(ethylene glycol)-block-polylysine was engineered to disassemble and undergo charge shifting at pH 6.0, releasing the azithromycin-conjugated poly(amidoamine) dendrimer (PAMAM) nanoparticles [8]. This system could enhance penetration and localization of antibiotics in biofilms and effectively eradicate bacterial count compared to free AZM. In another representative study,  enzyme-degradable polymeric vesicles with β-lactamase (Bla)-cleavable side chain motifs in the hydrophobic block, selectively delivered vancomycin and inhibited growth of MRSA strains [10].”

Comment # 22

 what is the interaction mode between nanoparticles and bacterial membrane?

RESPONSE: The following sentences are clarifying this issue in the manuscript:

“Chitosan-based nanocarrier properties can be controlled by modifying side chain hydrophobicity to expose the positively charged moieties on the surface. These positively charged chitosan carriers can bind with bacteria, with a negatively charged membrane surface, to promote membrane disruption and intracellular delivery of drugs.”

“Metal ions or metal nanoparticles interact directly with bacteria via various modes of action including membrane disruption (initiated by electrostatic interaction of positively charged silver ions with the negatively charged bacterial cell membrane), internal biomolecule damage, and the formation of reactive oxygen species [32].

Comment # 23

There should be a section about antibacterial NPs against biofilms forming bacteria, intracellular bacteria, etc.

RESPONSE: This review subsections are categorized based on the activity enhancer categories used for antibiotic delivery and therefore antibacterial activity of nanoparticles against biofilms is not presented in the separate section (antibacterial nanoparticles against biofilms forming bacteria, intracellular bacteria etc. are discussed in variety sections of the manuscript).

Comment # 24

There are many recent reviews similar to this topic and I can't see the novelty and the new added value in this review (https://doi.org/10.3390/pharmaceutics13111795https://doi.org/10.1080/14787210.2021.1908125; doi: 10.3390/nano10030560) and many more.

RESPONSE: The published reviews were focused on antimicrobial peptides (excluding all other antimicrobial compounds) or on specific delivery systems (e.g. metal-based), while our review has not such limitations. However, to acknowledge presence of above articles we have added them into the reference list. As reference number 25, 32, and 41.

Reviewer 3 Report

1)A detailed list of abbreviations could be tabulated at the start of the manuscripts. There are many abbreviations used in the manuscript.

2) Chapter 3.2: this chapter can be expanded to highlight on the importance of this delivery system.

3) Chapter 4: it is mentioned that silver and zinc metals are the most reported antimicrobial agents in literature, however, there are limited examples tabulated in table 1. Are the selected/quoted silver and zinc metals references good representative of the antibacterial potency of metal nanoparticles complexes with antibiotics.

4) Chapter 4.2 and 4.3: this chapter is too brief and could be expanded to highlight the significance of mentioned metals.

5) References: Formatting to be check carefully.

Overall, the draft is informative, but still lacking of information, examples, details, etc… for a good well-researched and conducive review article. I therefore suggest minor REVISION.

Author Response

Reviewer # 3

Comment # 1

A detailed list of abbreviations could be tabulated at the start of the manuscripts. There are many abbreviations used in the manuscript.

RESPONSE: the abbreviations were added as requested.

AMR

Antimicrobial resistance

MDR

Multidrug-resistance

AMP

Antimicrobial peptide

MRSA

Methicin-resistant Staphyloccocus aureus

ROS

Reactive oxygen species

PLGA

Poly lactic-glycolic acid

PEG

 Polyethylene glycol

SLN

Solid lipid nanoparticles

NLC

Nanostructured lipid carriers;

SLPN

Solid lipid-polymer hybrid nanoparticles

DOX

Doxycycline

TIMS,

Tilmicosin

SDS

Sodium dodecyl sulphate

PVP

Polyvinylpyrrolidone

SOS

Save our ship

rhBMP-2

recombinant human bone morphogenetic protein-2

HACC

Hydroxypropyltrimethyl ammonium chloride derivatized chitosan

MIC

Minimum inhibitory concentration

MBC

Minimum bactericidal concentration

ZOI

Zone of inhibition

LCHNPs

lipid-coated hybrid nanoparticles

GSH

Glutathione

PAMAM

Poly(amidoamine)

PAM

4-aminophenyl-alpha-d–manno-pyranoside

PEG-FeNPs

Polyethylene glycol iron oxide nanoparticles

Comment # 2

Chapter 3.2: this chapter can be expanded to highlight the importance of this delivery system.

Thank you for your constructive comment. We extended more information related to their applications into this part of the manuscript as follow:

“A series of nanoemulsions containing either cationic, non-ionic surfactants or their blended components were formulated together with ethylenediaminetetraacetic acid in the aqueous phase. The combination of cationic/non-ionic surfactants provided the greatest nanoemulsion stability. This formulation (named NB-201) was more effective against S. aureus compared to a commercially available silver sulfadiazine solution in the infected wound mouse model [29]. Additionally, the topical treatment of NB-201 inhibited the bacterial load in MRSA infected wounds in both in vitro and in vivo murine and porcine models. NB-201 was able to significantly reduce the inflammatory cytokine secretions induced by MRSA infection [30]. Moreover, the study of Cecchini et. al demonstrated that the formulation of nanoemulsion based on Minthostachys verticillata (Griseb.) eessential oil inhibited the growth of S. aureus without causing any toxicity to porcine and equine red blood cells [31].

Comment # 3

Chapter 4: it is mentioned that silver and zinc metals are the most reported antimicrobial agents in literature, however, there are limited examples tabulated in table 1. Are the selected/quoted silver and zinc metals references good representative of the antibacterial potency of metal nanoparticles complexed with antibiotics.

RESPONSE:  We thank the reviewer for pointing out the limitation of this section, but the intention of the review is to describe the representative “delivery systems or/and activity enhancers“, and metal components are mentioned as enhancers of antibiotics potency and the section about them is already significantly expanded comparing to other part of this review article, therefore we think that for the purpose of this review the selected silver and zinc metals are good (recently reported) representative examples.

Comment # 4

Chapter 4.2 and 4.3: this chapter is too brief and could be expanded to highlight the significance of mentioned metals.

RESPONSE:  Please refer to response to the comment 3. In addition, the section about them is already significantly expanded comparing to other part of this review article, therefore here are only mentioned the most significant examples and for readers interested in more details two recently published reviews about the antimicrobial potency of metals, their complexes and synergy with antibiotics was added to the text (reference 217 and 218).

Comment # 5

References: Formatting to be checked carefully.

RESPONSE: It has been checked as requested.

Overall, the draft is informative, but still lacking in information, examples, details, etc… for a good well-researched, and conducive review article. I, therefore, suggest minor REVISION.

RESPONSE: We have expanded some sections, added new examples, etc. as listed in answers to reviewer 2 and 3 comments.

Following references were added to the text:

  1. Blanchard, J.S. Molecular mechanisms of drug resistance in Mycobacterium tuberculosis. Annual review of biochemistry 1996, 65, 215-239.
  2. Arakha, M.; Pal, S.; Samantarrai, D.; Panigrahi, T.K.; Mallick, B.C.; Pramanik, K.; Mallick, B.; Jha, S. Antimicrobial activity of iron oxide nanoparticle upon modulation of nanoparticle-bacteria interface. Scientific reports 2015, 5, 1-12.
  3. Gurunathan, S.; Han, J.W.; Kwon, D.-N.; Kim, J.-H. Enhanced antibacterial and anti-biofilm activities of silver nanoparticles against Gram-negative and Gram-positive bacteria. Nanoscale research letters 2014, 9, 1-17.
  4. Weldrick, P.J.; Iveson, S.; Hardman, M.J.; Paunov, V.N. Breathing new life into old antibiotics: overcoming antibacterial resistance by antibiotic-loaded nanogel carriers with cationic surface functionality. Nanoscale 2019, 11, 10472-10485, doi:10.1039/c8nr10022e.
  5. Fux, C.A.; Costerton, J.W.; Stewart, P.S.; Stoodley, P. Survival strategies of infectious biofilms. Trends in Microbiology 2005, 13, 34-40, doi:10.1016/j.tim.2004.11.010.
  6. Komnatnyy, V.V.; Chiang, W.-C.; Tolker-Nielsen, T.; Givskov, M.; Nielsen, T.E. Bacteria-Triggered Release of Antimicrobial Agents. Angewandte Chemie-International Edition 2014, 53, 439-441, doi:10.1002/anie.201307975.
  7. Wu, Y.; Song, Z.; Wang, H.; Han, H. Endogenous stimulus-powered antibiotic release from nanoreactors for a combination therapy of bacterial infections. Nature Communications 2019, 10, doi:10.1038/s41467-019-12233-2.
  8. Gao, Y.; Wang, J.; Chai, M.; Li, X.; Deng, Y.; Jin, Q.; Ji, J. Size and Charge Adaptive Clustered Nanoparticles Targeting the Biofilm Microenvironment for Chronic Lung Infection Management. Acs Nano 2020, 14, 5686-5699, doi:10.1021/acsnano.0c00269.
  9. Huang, Y.; Ding, X.; Qi, Y.; Yu, B.; Xu, F.-J. Reduction-responsive multifunctional hyperbranched polyaminoglycosides with excellent antibacterial activity, biocompatibility and gene transfection capability. Biomaterials 2016, 106, 134-143, doi:10.1016/j.biomaterials.2016.08.025.
  10. Li, Y.; Liu, G.; Wang, X.; Hu, J.; Liu, S. Enzyme-Responsive Polymeric Vesicles for Bacterial-Strain-Selective Delivery of Antimicrobial Agents. Angewandte Chemie-International Edition 2016, 55, 1760-1764, doi:10.1002/anie.201509401.
  11. Su, F.-Y.; Chen, J.; Son, H.-N.; Kelly, A.M.; Convertine, A.J.; West, T.E.; Skerrett, S.J.; Ratner, D.M.; Stayton, P.S. Polymer-augmented liposomes enhancing antibiotic delivery against intracellular infections. Biomaterials Science 2018, 6, 1976-1985, doi:10.1039/C8BM00282G.
  12. Hamed, A.; Osman, R.; Al-Jamal, K.T.; Holayel, S.M.; Geneidi, A.-S. Enhanced antitubercular activity, alveolar deposition and macrophages uptake of mannosylated stable nanoliposomes. Journal of Drug Delivery Science and Technology 2019, 51, 513-523, doi:https://doi.org/10.1016/j.jddst.2019.03.032.
  13. Osman, N.; Devnarain, N.; Omolo, C.A.; Fasiku, V.; Jaglal, Y.; Govender, T. Surface modification of nano-drug delivery systems for enhancing antibiotic delivery and activity. Wiley Interdiscip Rev Nanomed Nanobiotechnol 2022, 14, e1758, doi:10.1002/wnan.1758.
  14. Bose, R.J.C.; Tharmalingam, N.; Choi, Y.; Madheswaran, T.; Paulmurugan, R.; McCarthy, J.R.; Lee, S.-H.; Park, H. Combating Intracellular Pathogens with Nanohybrid-Facilitated Antibiotic Delivery. International journal of nanomedicine 2020, 15, 8437-8449, doi:10.2147/IJN.S271850.
  15. Tan, C.H.; Jiang, L.; Li, W.; Chan, S.H.; Baek, J.-S.; Ng, N.K.J.; Sailov, T.; Kharel, S.; Chong, K.K.L.; Loo, S.C.J. Lipid-Polymer Hybrid Nanoparticles Enhance the Potency of Ampicillin against Enterococcus faecalis in a Protozoa Infection Model. ACS infectious diseases 2021, 7, 1607-1618, doi:10.1021/acsinfecdis.0c00774.
  16. Li, Y.; Liu, Y.; Ren, Y.; Su, L.; Li, A.; An, Y.; Rotello, V.; Zhang, Z.; Wang, Y.; Liu, Y.; et al. Coating of a Novel Antimicrobial Nanoparticle with a Macrophage Membrane for the Selective Entry into Infected Macrophages and Killing of Intracellular Staphylococci. Advanced Functional Materials 2020, 30, 2004942, doi:https://doi.org/10.1002/adfm.202004942.
  17. Fan, Y.; Marioli, M.; Zhang, K. Analytical characterization of liposomes and other lipid nanoparticles for drug delivery. Journal of Pharmaceutical and Biomedical Analysis 2021, 192, 113642, doi:https://doi.org/10.1016/j.jpba.2020.113642.
  18. Sharma, A.; Sharma, U.S. Liposomes in drug delivery: Progress and limitations. International Journal of Pharmaceutics 1997, 154, 123-140, doi:https://doi.org/10.1016/S0378-5173(97)00135-X.
  19. Silva, A.C.; Santos, D.; Ferreira, D.; Lopes, C.M. Lipid-based nanocarriers as an alternative for oral delivery of poorly water- soluble drugs: peroral and mucosal routes. Curr Med Chem 2012, 19, 4495-4510, doi:10.2174/092986712803251584.
  20. Cheaburu-Yilmaz, C.N.; Karasulu, H.Y.; Yilmaz, O. Chapter 13 - Nanoscaled Dispersed Systems Used in Drug-Delivery Applications. In Polymeric Nanomaterials in Nanotherapeutics, Vasile, C., Ed.; Elsevier: 2019; pp. 437-468.
  21. Khan, O.; Chaudary, N. The Use of Amikacin Liposome Inhalation Suspension (Arikayce) in the Treatment of Refractory Nontuberculous Mycobacterial Lung Disease in Adults. Drug design, development and therapy 2020, 14, 2287-2294, doi:10.2147/DDDT.S146111.
  22. Jiang, L.; Lee, H.W.; Loo, S.C.J. Therapeutic lipid-coated hybrid nanoparticles against bacterial infections. RSC Advances 2020, 10, 8497-8517, doi:10.1039/C9RA10921H.
  23. Aboumanei, M.H.; Mahmoud, A.F.; Motaleb, M.A. Formulation of chitosan coated nanoliposomes for the oral delivery of colistin sulfate: in vitro characterization, (99m)Tc-radiolabeling and in vivo biodistribution studies. Drug Dev Ind Pharm 2021, 47, 626-635, doi:10.1080/03639045.2021.1908334.
  24. Laverde-Rojas, V.; Liscano, Y.; Rivera-Sánchez, S.P.; Ocampo-Ibáñez, I.D.; Betancourt, Y.; Alhajj, M.J.; Yarce, C.J.; Salamanca, C.H.; Oñate-Garzón, J. Antimicrobial Contribution of Chitosan Surface-Modified Nanoliposomes Combined with Colistin against Sensitive and Colistin-Resistant Clinical Pseudomonas aeruginosa. Pharmaceutics 2020, 13, doi:10.3390/pharmaceutics13010041.
  25. Li, Y.; Tang, C.; Zhang, E.; Yang, L. Electrostatically entrapped colistin liposomes for the treatment of Pseudomonas aeruginosa infection. Pharm Dev Technol 2017, 22, 436-444, doi:10.1080/10837450.2016.1228666.
  26. Faya, M.; Hazzah, H.; Andeve, C.; Agrawal, N.; Maji, R.; Walvekar, P.; Albericio, F.; Govender, T. Novel formulation of antimicrobial peptides enhances antimicrobial activity against methicillin-resistant Staphylococcus aureus (MRSA). Amino Acids 2020, 52, doi:10.1007/s00726-020-02903-7.
  27. Ron-Doitch, S.; Sawodny, B.; Kühbacher, A.; David, M.M.N.; Samanta, A.; Phopase, J.; Burger-Kentischer, A.; Griffith, M.; Golomb, G.; Rupp, S. Reduced cytotoxicity and enhanced bioactivity of cationic antimicrobial peptides liposomes in cell cultures and 3D epidermis model against HSV. J Control Release 2016, 229, 163-171, doi:10.1016/j.jconrel.2016.03.025.
  28. Chauhan, M.K.; Bhatt, N. Bioavailability Enhancement of Polymyxin B With Novel Drug Delivery: Development and Optimization Using Quality-by-Design Approach. J Pharm Sci 2019, 108, 1521-1528, doi:10.1016/j.xphs.2018.11.032.
  29. Fan, Y.; Ciotti, S.; Cao, Z.; Eisma, R.; Baker, J., Jr.; Wang, S.H. Screening of Nanoemulsion Formulations and Identification of NB-201 as an Effective Topical Antimicrobial for Staphylococcus aureus in a Mouse Model of Infected Wounds. Mil Med 2016, 181, 259-264, doi:10.7205/milmed-d-15-00186.
  30. Cao, Z.; Spilker, T.; Fan, Y.; Kalikin, L.M.; Ciotti, S.; LiPuma, J.J.; Makidon, P.E.; Wilkinson, J.E.; Baker, J.R., Jr.; Wang, S.H. Nanoemulsion is an effective antimicrobial for methicillin-resistant Staphylococcus aureus in infected wounds. Nanomedicine (Lond) 2017, 12, 1177-1185, doi:10.2217/nnm-2017-0025.
  31. Cecchini, M.E.; Paoloni, C.; Campra, N.; Picco, N.; Grosso, M.C.; Soriano Perez, M.L.; Alustiza, F.; Cariddi, N.; Bellingeri, R. Nanoemulsion of Minthostachys verticillata essential oil. In-vitro evaluation of its antibacterial activity. Heliyon 2021, 7, e05896, doi:https://doi.org/10.1016/j.heliyon.2021.e05896.
  32. Godoy-Gallardo, M.; Eckhard, U.; Delgado, L.M.; de Roo Puente, Y.J.D.; Hoyos-Nogués, M.; Gil, F.J.; Perez, R.A. Antibacterial approaches in tissue engineering using metal ions and nanoparticles: From mechanisms to applications. Bioactive Materials 2021, 6, 4470-4490, doi:https://doi.org/10.1016/j.bioactmat.2021.04.033.

Round 2

Reviewer 2 Report

No Further comments to the authors